# Flow Cytometry of Oxygen and Oxygen-Related Cellular Stress

Beatriz Jávega [1] , Guadalupe Herrera [2], Alicia Martínez-Romero [3] and José-Enrique O'Connor [1,*]

1    Laboratory of Cytomics, Joint Research Unit CIPF-UVEG, The University of Valencia and Principe Felipe Research Center, 46012 Valencia, Spain; beatriz.javega@uv.es
2    Cytometry Service, Central Research Unit (UCIM), Incliva Foundation, The University of Valencia and University Clinical Hospital, 46010 Valencia, Spain; guadalupe.herrera@uv.es
3    Cytomics Technological Service, Principe Felipe Research Center, 46012 Valencia, Spain; amartinez@cipf.es
*    Correspondence: jose.e.oconnor@uv.es; Tel.: +34-963-86-49-88

**Abstract:** Reactive oxygen species (ROS) are unstable and highly reactive molecular forms that play physiological roles in cell signaling and immune defense. However, when ROS generation is not properly balanced by antioxidant defenses, a pathological condition known as oxidative stress arises, in association with the onset and progression of many diseases and conditions, including degeneration and aging. Biomarkers of oxidative stress in biomedicine are actively investigated using different approaches, among which flow cytometry (FCM) and other single-cell, fluorescence-based techniques are most frequent. FCM is an analytical method that measures light scattering and emission of multiple fluorescences by single cells or microscopic particles at a very fast rate. To assess the specific role of ROS in oxidative stress, it is essential to detect and characterize these species accurately. However, the detection and quantitation of individual intracellular ROS and parameters of oxidative stress using fluorogenic substrates and fluorescent probes are still a challenge, because of biological and methodological issues. In this review, we present and discuss a series of complementary strategies to detect ROS or to focus on other endpoints of oxidative stress. Based on our results, we propose some recommendations for proper design of cytometric studies of oxidative stress in order to prevent or minimize the limitations and experimental errors of such approaches.

**Keywords:** fluorescence; image cytometry; cytomics; reactive oxygen species; antioxidants; oxidative stress; in vitro toxicology





## 1. Introduction: Oxygen, ROS and Oxidative Stress

Most living organisms depend on oxygen to obtain metabolic energy from the oxidation of biomolecules [1]. However, the oxygen functions essential to living things depend on a chemical property dangerous to them: the structure of the oxygen molecule ($O_2$) has two unpaired electrons, and $O_2$ can accept individual electrons to generate unstable and highly reactive molecular forms, known as reactive oxygen species (ROS) [2].

The term ROS refers to both free radicals and molecules derived from them [3]. In fact, $O_2$ itself is a free radical, of relatively low reactivity [2]. More active, biologically relevant ROS derived from metabolic or abiotic reactions of $O_2$ include singlet oxygen ($^1O_2$), superoxide anion radical ($O_2^{\bullet-}$), hydrogen peroxide ($H_2O_2$) and hydroxyl radical ($OH^\bullet$) [1,2].

$^1O_2$ is an excited and much more reactive form of the $O_2$ molecule, that can arise by photochemical reactions or by abiotic stress [4]. The pathway of univalent $O_2$ reduction, as it occurs along the mitochondrial respiratory chain, generates $O_2^{\bullet-}$, a relatively unreactive species that can undergo enzyme- or metal-catalyzed processes involving many molecules [2,5] and eventually produce $H_2O_2$ and other ROS. $H_2O_2$ is not a free radical, but its relatively long half-life and its capacity for diffusing easily through cell membranes [5] enable $H_2O_2$ to undergo metal-catalyzed reactions, yielding the very strong $^\bullet OH$ radical

that reacts readily with cellular macromolecules, including DNA, proteins, lipids and sugars, thus being one of the most potentially damaging ROS [2,5].

There are also nitrogen-containing free radicals and reactive molecules, the reactive nitrogen species (RNS), including nitric oxide (NO) and peroxynitrite (ONOO) [5,6]. Because RNS contain oxygen and RNS and ROS biochemistry are closely related, ROS and RNS have been grouped together under the term reactive oxygen and nitrogen species (RONS) [7].

RONS have been involved in aging and cellular senescence [8], as well as in diverse inborn [9,10] or acquired conditions, such as inflammatory [11–14] and cardiovascular diseases [15–18], thrombosis [19], cancerogenesis [20–23] and, paradoxically, anticancer chemotherapy [24], viral infections [25,26], neurodegenerative conditions [27–29] and metabolic syndromes [30]. On the other hand, RONS are partners in multiple intercellular- and intracellular signaling [31–34] and play important roles in homeostatic processes involved both in the destruction of invading pathogens [35] or in the adaptation to endogenous and exogenous stress [36–38].

The physiological levels of RONS are controlled by a complex system of antioxidant processes. However, when the efficiency of such mechanisms is overcome by the intensity or the duration of oxidative reactions, oxidative stress ensues, a situation which can be defined as an alteration in the balance between the generation of ROS production and the antioxidant defenses, leading to oxidative damage [39,40]. Oxidative stress may be caused by two separate processes, that are not excluding. On the one hand, the ROS themselves may decrease the level or activity of antioxidant enzymes by mutation or destruction of their active center [40]. On the other hand, increased production of ROS, exposure of cells or organisms to elevated levels of exogenous ROS or their metabolic precursors, and even excessive induction of ROS-mediated protective processes, such as phagocytosis or xenobiotic biotransformation can lead to oxidative stress [2,5].

Biomarkers of oxidative stress in biomedicine are actively investigated using many different methods and approaches [41], in order to assess the redox state and its alterations in the whole organisms or in specific tissues and cell types [42–44].

Fluorescence is an advantageous methodology because of its simplicity, its high sensitivity and the spatial resolution of modern imaging techniques [45,46]. For these reasons, fluorescent probes are used extensively for investigating the participation of ROS in oxidative stress, both in experimental- or clinical settings [45,46], often involving the use of flow cytometry (FCM) [47,48] and related single-cell-based technologies [49,50].

However, many issues complicate the fluorescence-based detection of ROS, including the low concentration of ROS, their short half-life and the extensive interactions among them, as well as the competing effect of the exogenous fluorescent probes with the physiological in situ reactions. All these aspects contribute to the intrinsic limitations of both probes and experimental conditions [51–53], and they should be taken into consideration. Moreover, the efficiency and specificity of many probes for detecting ROS in vitro have not been unequivocally established [51–53]. Such limitations and potential sources of artifacts challenge the accuracy quantitative measurements of intracellular generation of ROS and demand a careful experimental design and a cautious interpretation of the results [53].

## 2. Flow Cytometry as a Tool for Functional Cell Research

FCM can be defined basically as an analytical method which measures light scattering and emission of multiple fluorescences by single cells or microscopic particles (natural or artificial) aligned by a laminar stream, when they are presented one-by-one at high speed to suitable illumination sources [54,55]. In a conventional flow cytometer, suspensions of cells, cell-derived elements or other relevant particles in an appropriate isotonic liquid medium are hydrodynamically focused across an illumination system and an optical system capable of simultaneously quantifying multiple structural and functional properties of each cell or particle. The characterization of cellular parameters at rates of thousands of events per

second is achieved mostly by fluorescence measurements or by other forms of interaction between light and matter [54,55].

FCM allows fast analysis of multiple simultaneous parameters in individual cells within a heterogeneous cell population, when transported by a liquid stream across an illumination source and light detectors. The computer-integrated data provide a comprehensive description of biological features of the sample. The growing Biomarkers of oxidative stress in biomedicine are actively investigated using different approaches availability of fluorescent reagents and the recent development of algorithms for multispectral-fluorescence unmixing allows quantifying up to 30–50 parameters at the same time [56]. The ability to identify cell subpopulations, including very rare cells, makes FCM an essential tool in Cellular and Molecular Biology, Biotechnology, Toxicology and Drug Discovery, or Environmental Studies [57]. On the other hand, FCM is a firmly established technology for clinical diagnosis and prognosis, especially in Immunology [58] and Onco-Hematology [59].

Fluorescence-based analysis of ROS and oxidative stress is a very relevant application of FCM as attested by the more than 8000 papers produced between 1989 and 2023, according to PubMed Central. However, ascertaining the specific role of ROS in oxidative stress studies by cytomic methodologies, require detecting and characterizing these species accurately. While the specific analysis of individual intracellular ROS remains a challenge [51–53], alternative cytometric approaches can be considered, aiming to describe other endpoints of oxidative stress. In this review, the limitations and perspectives of such approaches are introduced and briefly discussed.

### 2.1. Specific Features and Limitations of Functional FCM

Because of the multiparametric analysis (and physical separation) of single cells or particles at very fast rate, FCM can be considered as a particular method for biochemical analysis, with advantages over other conventional methodologies applied to the study of oxidative stress.

### 2.1.1. Multiparametric Data Acquisition

Most biochemical methods quantitate a single biomarker per assay, and are not suitable for single-cell analysis. On the contrary, FCM instruments routinely allow to examine two morphology-related parameters (forward and side-light scatter) and up to 30–50 fluorescence signals per single particle [59,60]. Accordingly, a single-tube assay, provides one or more parameters may be used to identify and select ("gated analysis") specific cell subsets within phenotypically- or functionally heterogeneous cell populations (e.g., live, apoptotic or necrotic cells; cells of different lineage; cells in different cell cycle phase), whereas other fluorescence signals may be reveal specific structures or specific functions in the individual elements of the selected subsets [54,55].

### 2.1.2. Multivariate Data Analysis

Because of to the hardware structure and software design of current flow cytometers, multiparametric acquisition is interfaced to multivariate data analysis. In this way, cell populations can be described by the multidimensional correlation of the individual properties measured on each single cell, thus enhancing the discriminating power [58,60]. Moreover, the special format in which FCM raw data and experimental details are stored as listmode files, i.e., uncorrelated data matrices for each cell (flow cytometry standard, FCS), allows to define post hoc new parameter correlations and population-gating criteria when replaying those FCS files [58,61]. This is an invaluable tool especially when scarce or rare samples are studied.

### 2.1.3. Fast Analysis of Large Number of Live Cells

FCM can be applied to study an ample diversity of cell types and samples, in different conditions of vitality (e.g., fresh, fixed and permeabilized cells) [58]. Live-cell analysis allows to determine biological processes in minimally perturbed intracellular environments

and in near-physiological conditions. The fast rate of analysis makes possible to examine in a reasonable time a large number of individual cells, allowing the accurate detection and analysis of infrequent events ("rare cells"), down to 1 event per $10^8$ cells [58]. This is in contrast with bulk biomolecular determinations in which components extracted from a collectivity of cells are analyzed, thus providing average values of gene expression, enzyme activity or metabolite concentration.

### 2.1.4. Real-Time Flow Cytometry

Including time as an operative parameter in FCM has led to the concept of real-time flow cytometry (RT-FCM), an experimental strategy in which the changes in cellular parameters within heterogeneous cell populations are followed in a kinetic fashion [47,62]. In RT-FCM, the biological process of interest progress as the sample is run in the flow cytometer and single cells are analyzed in sequence. RT-FCM allows to monitor functional changes in high numbers of single cells with a theoretical time-resolution of less than one second, and for time windows ranging from few seconds to several minutes. This type of kinetic analysis is specially relevant for following very fast or transient dynamic processes, as those typical of signal transduction [62].

### 2.1.5. Individual Cell Sorting

FCM can not only analyze different cell populations based on the use of fluorescent probes, but can also separate these populations using the same cytometric detection principles. Cell sorting allows the combination of the intrinsic capabilities of FCM results with information obtained by image analysis and molecular techniques and provides a preparative tool for rapid isolation of rare cells of biochemical relevance [63].

The most frequent and complex cell sorters are based on the formation and deflection of microdrops by high-frequency vibration, and charged electric plates to deflect these droplets towards specific collection tubes or supports. In general, this type of separator allows several sub-populations to be separated simultaneously and at a very high speed (thousands of cells per second). A second family of cell separators are based on microfluidics. These systems, in general, require a smaller number of starting cells, but they are of lower speed and performance than electrostatic separators. On the other hand, they present fewer biological risks for the operator and are less aggressive towards the cells in process [58].

### 2.1.6. Limitations of Functional Flow Cytometry

While the large number of cells analyzed and the instrumental settings of current cytometers provide multiple strategies to obtain primary information, and allow a large number of general applications, there are several critical points and difficulties when performing adequate functional analysis by FCM (Table 1). The limitations depend mostly on the maintenance of adequate viability or metabolic capacity of cells along sample preparation and analysis, as well as avoiding the interference of fluorescent probes with cellular functions [53].

**Table 1.** Summary of main critical points and limitations of flow cytometry in functional assays of oxygen and oxygen-related stress.

| Critical Points and Limitations |
| --- |
| Identification of blood cells in whole-blood samples without lysis of erythrocytes |
| Preparation of single-cell suspensions from adherent cell models |
| Maintenance of viability and functional competence of the cells along sample preparation and experiment performance |
| Identification of the optimal incubation time and concentration for staining |
| Access of fluorogenic substrates to intracellular sites or intracellular processes |

**Table 1.** *Cont.*

| Critical Points and Limitations |
| --- |
| Retention of fluorogenic substrates and oxidized fluorescent probes |
| Preparation of single-cell suspensions from adherent cell models |
| Lack of absolute specificity of fluorogenic substrates for specific RONS |
| Interference of the probes with ROS biology or ROS-relevant cell functions |
| Selection of the time window for kinetic assays |
| Assay calibration for data expression in biochemical units |
| Adapted from [53] |

**3. General Strategies in Flow Cytometric Analysis of Oxygen and Oxidative Stress**

Possibly the most frequent and standardized application of FCM in the field of oxidative stress is the ex vivo assessment of the phagocytic respiratory for the diagnosis or prognosis of chronic granulomatous disease and sepsis [58,64]. In the areas of Cell Biology and Biotechnology, the most relevant examples of FCM applications for **in vitro** research of oxidative stress can be found in the assessment of the involvement of ROS in physiopathological conditions [64–69], the evaluation of the mechanisms involved in xenobiotic biotransformation and toxicity [49,69], and the analysis of antioxidant properties of drugs and natural compounds [66,69,70].

FCM is a versatile technology that provides different approaches to study the complex mechanisms involved in the biochemistry of ROS, their control by antioxidant systems and the consequences of their action. Thus, the most common FCM strategies for the study of oxygen, ROS and oxidative stress include:

Performing cell-based studies in hypoxic conditions:

Hypoxia can be defined as oxygen deficiency of the cellular environment and hypoxic responses mediated by hypoxia-inducible (HIF) transcription factors are associated with several pathologies, including lipid metabolism, inflammation, cardiovascular disease, hypertension, tumor-mediated immunosuppression, and neurodegenerative disease [2,71]. Hypoxia is a key factor in primary tumors and metastasis, including cell proliferation, metabolic capacity, immune response, and drug resistance to chemotherapeutic intervention [72].

FCM is commonly applied in cell-based studies comparing experimental conditions of normoxia versus hypoxia, mostly related to stem-cell [73–75] and cancer research [76,77] as well as for assessing oxidative stress [78], mitochondrial function [79,80] and angiogenesis [81] associated to the hypoxia/reoxygenation transition.

Monitoring intracellular oxygen in hypoxic conditions:

Hypoxia can be monitored in living tissues with complex methods, including, magnetic resonance imaging (MRI), positron-emission tomography (PET) and computerized tomography (CT) [82]. Such methods have shortcomings and cannot be applied to the single-cell level or to finely detect local heterogeneity in hypoxia. On the contrary, a series of small fluorescent probes for hypoxia have clear advantages of simplicity, sensitivity, and high temporal- and topological resolution [83]. The mechanism of such probes is based on the hypoxia-induced overexpression of reductases catalyzing oxygen-sensitive bioreductive reactions, such as cytochrome P450 reductase and nitroreductase [83,84].

Direct detection of ROS, the initiators of the oxidative stress process:

This application is complicated due to the low concentration, short half-life and multiple interactions of ROS, as well as by the limitations imposed by the fluorescent probes and the experimental conditions [53]. As schematically shown in Figures 1 and 2, typically, the fluorescent probes are non fluorescent until being oxidized by intracellular oxidants and they are incorporated in form of fluorogenic substrates (Figure 1) which have been

modified by appropriate chemical design to become both cell-permeable and susceptible to ROS-mediated oxidation (Figure 2) [51–53]. Figure 3 exemplifies the application of FCM to perform endpoint and real-time analysis of ROS generation.

Detection of more stable oxidized end products:

FCM can be applied to the analysis of stable products generated by the reaction of RONS with endogenous molecules or with exogenous probes added to this purpose. This strategy includes the detection of lipid-peroxidation products and of ROS-induced damage to DNA [2,53].

Assessment of antioxidant defenses, mostly GSH and SH-containing proteins:

This indirect cytometric approach to study oxidative stress may be limited by issues related to the complexity of the antioxidant defense by itself and to the specificity of enzymes required to fluorescent reporting of the process [2,53].

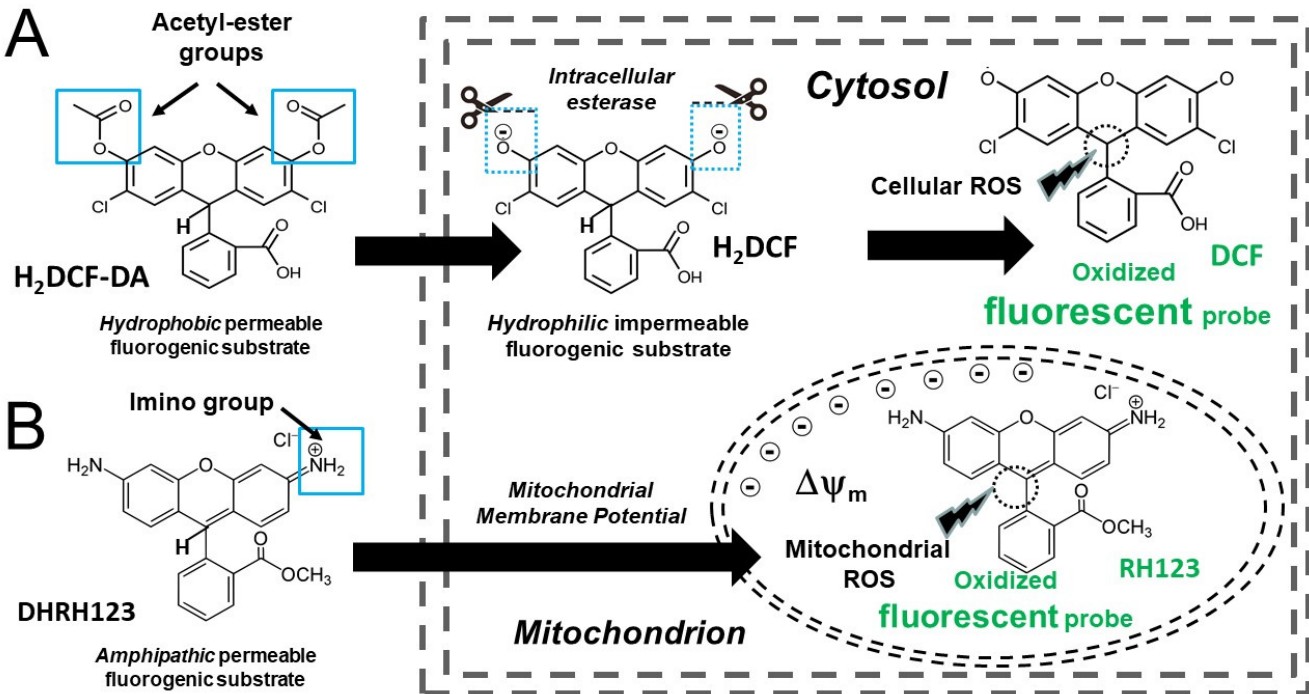

**Figure 1.** Example of biochemical mechanisms involved in the localization and generation of intracellular ROS-sensitive fluorescent probes from extracellular fluorogenic substrates. (**A**) The incorporation of two acetyl-ester groups renders the chemically reduced, non-fluorescent 2′,7′-Dichlorodihydrofluorescein diacetate (H$_2$DCF-DA) molecule totally non-polar and, thus, able to cross freely plasma membrane. Intracellular esterases cleave acetyl groups and expose two negative charges, thus retaining this substrate in the cytosolic compartment. Intracellular ROS oxidize the molecule and generate a fluorescent probe. This general mechanism is extensible to other esterified fluorogenic substrates. (**B**) Because of positively charged imino groups (=NH$_2$), dihydrorhodamine 123 (DHRH123) is an amphipathic, non-fluorescent fluorogenic substrate permeable to the plasma membrane. Intracellular DHRH123 is transported into mitochondria driven by the mitochondrial membrane potential that reflects the proton-motive force ($\Delta\psi$m) generated by the respiratory electron transfer chain. Once in the mitochondrial matrix, DHRH123 is oxidized by intramitochondrial ROS to the fluorescent probe rhodamine 123 (RH123). A similar mechanism applies for other cationic fluorogenic substrates.

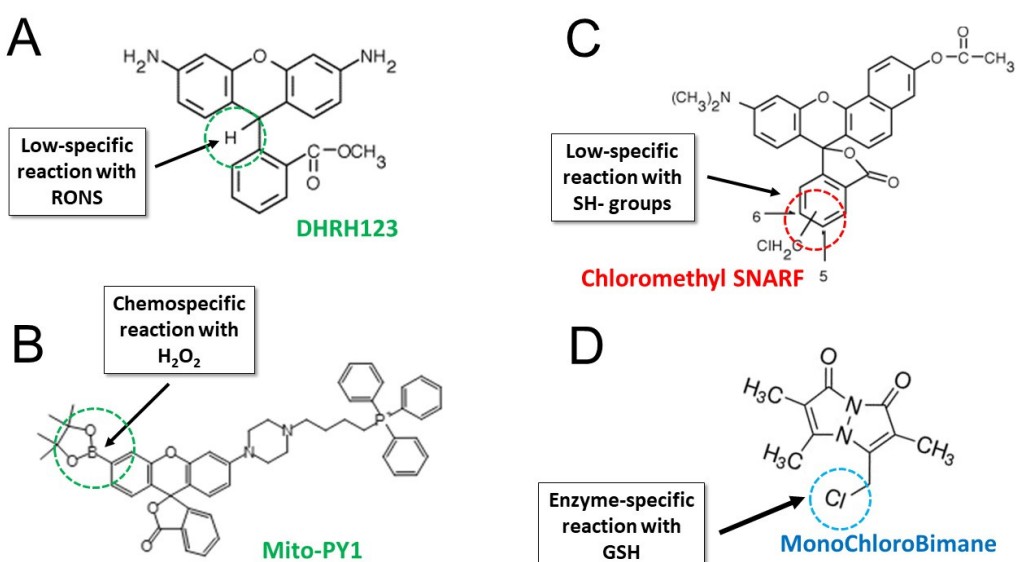

**Figure 2.** Examples of specificity issues in common fluorescent probes used for detecting ROS in mitochondria (**A**,**B**) and for assessing Glutathione (GSH)-related antioxidant activity. DHRH123 (**A**) and Mito-PY1 (**B**) are both used for detection of intramitochondrial ROS. Both probes share the same electrochemical mechanism for intramitochondrial localization, as described in Figure 1 for DHRH123. However, several ROS and RNS can react with the indicated H- atom in DHRH123, while the boronate group in Mito-PY1 undergoes a highly chemospecific reaction only with $H_2O_2$. Chloromethyl SNARF (**C**) and monochloro-bimane (**D**) are commonly used for GSH detection. However, because of its strong reactivity, the chloromethyl group in chloromethyl SNARF forms adducts with most thiol (-SH) groups. On the contrary, monochloro-bimane is a substrate of the enzyme GSH-transferase (GST) and forms adducts only with GSH.

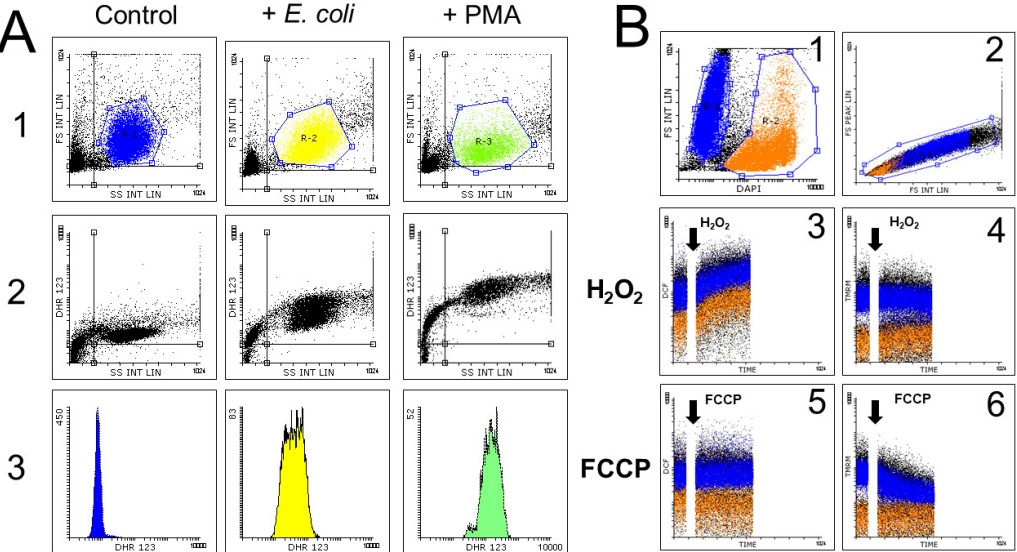

**Figure 3.** Representative examples of (**A**) endpoint and (**B**) real-time kinetic flow cytometric analysis of ROS generation. Panel A summarizes graphically the performance of the oxidative burst in whole-blood phagocytes of a healthy donor, using a commercial kit for clinical diagnosis of the chronic granulomatous disease (CGD) (FagoFlowex) under manufacturer's instructions. Samples of human peripheral whole blood are incubated for 30 min with the fluorogenic substrate DHRH123 in the presence of vehicle (column Control) or Escherichia coli (column + *E. coli*). For positive control (column + PMA), blood samples are incubated with the phorbol ester PMA, an activator of Protein kinase C (PKC). Row A1 shows the morphological identification and gating of phagocytic cells in a

FS INT Lin vs. SS INT Lin dotplot. Row A2 displays the ROS generation (DHRH123 vs. SS INT Lin plots) during the oxidative burst induced by phagocytosis (+*E. coli*) or PKC activation (+PMA), as compared with the spontaneous ROS generation induced by the experimental conditions (Control). Row 3 compares the monoparametric histograms of fluorescence intensity of ROS-generated RH123 in phagocytes of the three experimental conditions. Such histograms are clinically used to diagnose CGD or other oxidative-burst related genetic defects [58]. Panel B exemplifies graphically the performance of a flow cytometry kinetic analysis (RT-FCM or In Fluxo analysis [62]) of the ROS generation in Jurkat cells treated with exogenous $H_2O_2$ or with the mitochondrial uncoupler FCCP [53]. Graph B1 shows the identification and gating of live (blue) and dead (orange) cells using FS signal and the viability stain DAPI [53] and graph B2 displays the gate used for single-cell selection in a FS Peak vs. FS INT dotplot. For In Fluxo experiments, Jurkat cells are incubated with the appropiate fluorogenic substrates and the viabiity marker. Dotplots display real time on *x*-axis and the fluorescence intensity of each probe in the *y*-axis (log scale). Cells are run in the cytometer for some seconds to establish baseline fluorescence, the run is halted for addition of the exogenous challenge and immediately continued until the stop time. Graph B3 shows that $H_2O_2$ addition to the cells results in a fast generation of intracellular ROS (measured with ROS-generated DCF from the substrate $H_2$DCF-DA), especially in the living cells (blue). Graph B4 shows that mitochondrial membrane potential (MMP) is slightly decreased by $H_2O_2$ addition in live cells (blue), but not in dead cells (orange), as indicated by the fluorescence of the mitochondrial probe tetramethyl rhodamine (TMRM), the product of the fluorogenic substrate tetramethyl rhodamine methyl ester (TMRM-ME). The effect of the mitochondrial uncoupler FCCP is shown in graphs B5 and B6. Mitochondrial uncoupling by FCCP induces a slight generation of ROS in live cells, but not in dead ones (graph B5), while MMP is strongly decreased by FCCP in live cells (graph B6). As seen in this plot, dead cells are characterized by an already very low MMP, which is not modified by FCCP.

### 3.1. Monitoring Intracellular Oxygen in Hypoxic Conditions

Classical hypoxia-sensing probes, such as pimonidazole and 2-(2-nitro-l-H-imidazol-1-yl)-N-(2,2,3,3,3-pentaIluoropropyl)-acetamide (EF5) [84,85], contain nitroimidazole groups that are reduced by reductases in hypoxia to generate covalently-bound products that require cell fixation followed by immunostaining to be detected. While such probes are currently applied to FCM studies of hypoxia [86–88], cell-permeant fluorogenic substrates have been developed that can be used in live single cells without the need of fixation/permeabilizaton procedures. Hypoxia Green Reagent™ for flow cytometry (λ excitation = 488 nm; λ emission = 530 nm) [89] and its related BioTracker 520 Green Hypoxia Dye™ [90] are suitable for monitoring hypoxia in living cells. Both probes are as sensitive as pimonidazole and may be used for live-cell fluorescence microscopy and FCM, with common spectral properties (λ excitation = 498 nm; λ emission = 520 nm). As cellular oxygen levels decrease, these probes responds by releasing rhodamine, which results in detectable emissions in the green channel.

iT™ Green Hypoxia Reagent (λ excitation = 488 nm; λ emission = 520 nm) and Image-iT™ Red Hypoxia Reagent (λ excitation = 490 nm; λ emission = 610 nm) are live-cell permeable compounds which increase fluorescence in environments with low oxygen concentrations. Unlike pimonidazole adducts, which only respond to oxygen levels lower than 1%, Image-iT™ reagents are fluorogenic when atmospheric oxygen levels are lower than 5%, and their fluorogenic response increases as the oxygen levels decrease in the environment [91,92].

Recently synthesized indolequinone-based bioreductive fluorescent probes allow imaging different levels of hypoxia in cell cultures [83]. These probes are modified versions of the existing resorufin and Me-Tokyo Green fluorophores. The resorufin-related probe is fluorescent in conditions of 4% $O_2$ and lower, while the Me-Tokyo Green-related probe is only fluorescent in severe hypoxia conditions of 0.5% $O_2$ and less [83].

Green-fluorescent protein (EGFP) constructs under the control of hypoxia-induced promoters have been used clasically as hypoxia reporters by fluorescence imaging of single cells, allowing dye-free monitoring of hypoxic responses, despite the $O_2$ requirement

for EGFP-fluorophore formation [93]. More recently, a novel family of hypoxia sensors have been described, based on UnaG, a fluorescent protein from Japanese freshwater eel that does not rely on $O_2$ to adopt the fluorescent state [94,95]. The combination of UnaG with oxygen-sensitive fluorescent proteins has led to several reporters of hypoxia and reoxygenation that allow to record the dynamics of hypoxia in living cells [96].

Figure 4 shows the chemical structures of selected relevant fluorescent probes used to monitor intracellular oxygen in hypoxic conditions.

**Pimonidazole**

**2-(2-nitro-1-H-imidazol-1-yl)-N-(2,2,3,3,3-pentalluoropropyl)-acetamide (EF5)**

**BioTracker™ 520 Green Hypoxia Dye**

**Resorufin-based Indolequinone Red Hypoxia Dye**

**Me-Tokyo-based Indolequinone Green Hypoxia Dye**

**Figure 4.** Chemical structures of relevant fluorescent probes used to monitor intracellular oxygen in hypoxic conditions. For preparing this figure, we have used MolView, an open source chemical modeling package (http://molview.org/, accessed on 20 May 2023). The chemical structures have been retrieved directly via MolView from the PubChem Compounds database or sketched by us with MolView from the structures published in references [83–96]. The structures are identified by their chemical name or their tradename, with the abbreviation in parenthesis.

### 3.2. Direct Detection of ROS with Fluorogenic Substrates

Fluorescent probes and fluorogenic substrates provide a convenient approach for detecting and quantifying ROS generation in different cellular models. However, their application has many limitations and artifacts that will be discussed further on.

### 3.2.1. $^1O_2$ Probes

The fluorescent probes designed for $^1O_2$ take advantage of its chemical reactivity and combine a chemical $^1O_2$ trap and diene-containing fluorophores [97–100]. On this chemical motif, structural modifications of $^1O_2$ probes result in different optical properties. Thus, 1,3 diphenylisobenzofuran (DPBF) reacts with $^1O_2$ to form a nonfluorescent endoperoxide [99]. On the contrary, derivatives of DPBF substituted with phenanthrene (PPBF), pyrene (PyPBF) and 4-(diphenylamino) stilbene (StPBF) can acts as ratiometric probe for $^1O_2$ detection. These $^1O_2$ probes undergo significant red shift in their emission spectrum, as the conjugation goes from DBPF to StPBF [99].

9-[2-(3-Carboxy-9,10-diphenyl)anthryl]-6-hydroxy-3H-xanthen-3-one (DPAX) is a sensitive and efficient fluorescent probe for the study of $^1O_2$ which combines a fluorescein skeleton with DPA [100]. DPAX and its related probes emit very low fluorescence in aqueous solution but when bound to $^1O_2$, the corresponding endoperoxide (DPAX-EP) emits intensely [100]. DPAX and its related compounds show an excellent selectivity towards

$^1O_2$, as compared with other RONS [100]. The fluorescence intensity of DPAX can be stabilized of by incorporating electron withdrawing groups, such as Cl or F, at the 2- and 7- positions of the xanthene moiety, leading to generation of DPAX-2 (Cl derivative) and DPAX-3 (F-derivative) [100].

9-[2-(3-Carboxy-9,10-dimethyl)anthryl]-6-hydroxy-3H-xanthen-3-one (DMAX) reacts rapidly with $^1O_2$ with much greater sensitivity than DPAX. Both DMAX and its endoperoxide DMAX-EP have similar spectral properties (λ excitation = 492 nm; and λ emission = 515 nm), compatible with most FCM instruments., DMAX-EP is highly fluorescent whereas DMAX itself is practically non-fluorescent. Further, the hydrophobicity of DMAX is lower than that of DPAXs, making it suitable for assays in biological sample [100].

More recently, dansyl-based dansyl-2,2,5,5-tetramethyl-2,5-dihydro-1H-pyrrole (DanePy) [101] and Singlet Oxygen Sensor Green reagent® (SOSG) [98,102] fluorescent probes have been synthesized, both including an anthracene moiety (electron donor) that quenches the fluorescence of the fluorochrome (electron acceptor) through electron transfer [102]. When the anthracene moiety traps $^1O_2$, the resulting adduct stops being an electron donor and the fluorescence is recovered [103].

SOSG is currently the choice probe for detection of $^1O_2$, due to its claimed high selectivity and specificity to $^1O_2$ [98,102–104] This fluorescein-based dye probe upon reaction with $^1O_2$, generates SOSG endoperoxide (SOSG-EP) that emit green fluorescence (λ excitation = 504–508 nm; λ emission = 525–536 nm) [103,104]. SOSG has spectral properties similar to those of fluorescein, making this probe suitable for FCM instruments. However, SOSG presents some drawbacks, due mostly to unequal penetration in cells. In addition, SOSG may induce photosensitization, resulting from $^1O_2$ generation by the probe itself under exposure to both UV radiation and visible light [104]. While SOSG does not show an appreciable response to $O_2^{\bullet-}$ and $HO^{\cdot}$, its fluorescence emmision increases dose-dependently when exposed to gamma-rays or X-rays, in experimental conditions where formation of $^1O_2$ may be ruled out [105].

The modified $^1O_2$ indicator Aarhus Sensor Green (ASG) (a tetrafluoro-substituted fluorescein derivative bound covalently to a 9,10-diphenyl anthracene moiety) has spectral properties similar to SOSG, but without inducing photosensitization [106].

Figure 5 summarizes the chemical structures of some relevant fluorescent probes used to monitor $^1O_2$.

### 3.2.2. 2′,7′-Dichlorodihydrofluorescein Diacetate and Related Probes

2′,7′-Dichlorodihydrofluorescein diacetate (H$_2$DCF-DA) is one of the most used cell-permeant fluorogenic substrates for ROS detection [51,107,108]. After hydrolysis of acetate groups by intracellular esterases, ROS-mediated oxidation of intracellular 2,7-dichlorodihydrofluorescein (H$_2$DCF) yields fluorescent 2,7-dichlorofluorescein (DCF; λ excitation = 498 nm; λ emission = 522 nm). While repeatedly reported as a specific indicator of H$_2$O$_2$ [109], H$_2$DCF is also oxidized by other ROS, such as $OH^{\bullet}$ and peroxyl radicals, and also by RNS such as ONOO [51,107,108]. On the other hand, it seems well established that H$_2$DCF is not oxidized by $O_2^{\bullet-}$, hypochlorous acid or NO [107]. With these limitations, H$_2$DCF has been successfully applied to studies of oxidative burst in phagocytes [110,111] and to assess oxidative processes in different cell models [112–116].

Intracellular oxidation of H$_2$DCF in conditions of cell damage, is usually accompanied by leakage of the oxidized end-product, DCF. To enhance retention of the oxidized probe, analogs with better retention have been synthetized, such as carboxylated H$_2$DCF-DA (carboxy-H$_2$DCF-DA), which has two negative charges at physiological pH, and its di-(acetoxymethyl ester) [117]. The halogenated compounds 5-(and 6-)chloromethyl-2′,7′-dichlorodihydrofluorescein diacetate, acetyl ester (CM- H$_2$DCF -DA), and 5-(and 6-) carboxy-2′,7′-difluorodihydrofluorescein diacetate (carboxy-H$_2$DFF-DA) are much better retained by live cells and have been used for monitoring oxidative burst in clinical settings and in different experimental models of oxidative stress [117].

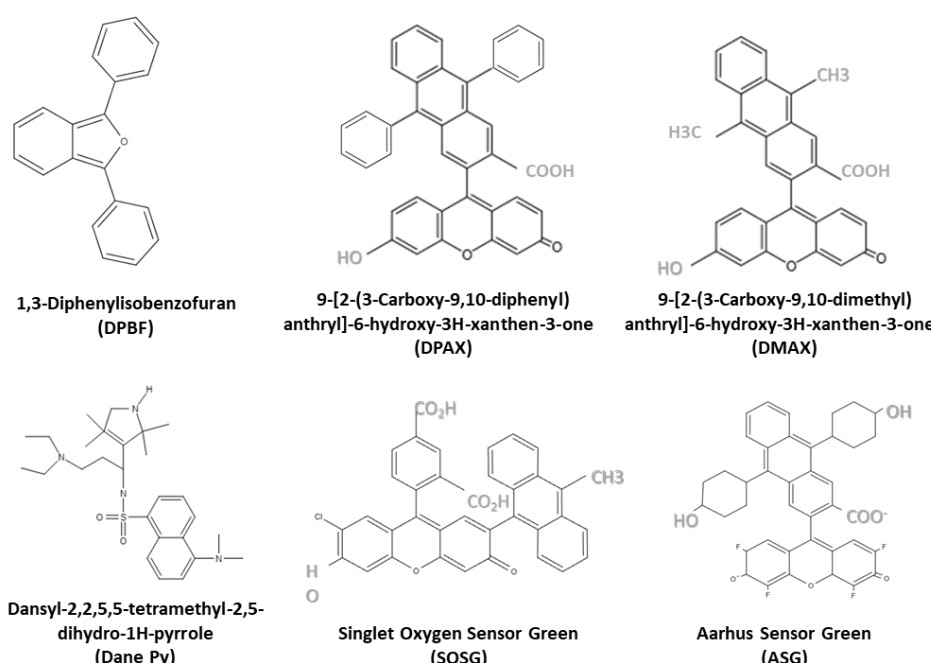

**Figure 5.** Chemical structures of relevant fluorescent probes used to monitor singlet oxigen. For preparing this figure, we have used MolView, an open source chemical modeling package (http://molview.org/, accessed on 20 May 2023). The chemical structures have been retrieved directly via MolView from the PubChem Compounds database or sketched by us with MolView from the structures published in references [97–106]. The structures are identified by their chemical name or their tradename, with the abbreviation in parenthesis.

### 3.2.3. Dihydrorhodamine 123

Dihydrorhodamine 123 (DHRH123) is a fluorogenic substrate that generates upon oxidation oxidation rhodamine 123, a cell-permeable fluorescent cationic probe ($\lambda$ excitation = 505 nm; $\lambda$ emission = 529 nm) [51,107]. Upon oxidation of DHRH123 to the fluorescent rhodamine 123, one of the two equivalent amino groups tautomerizes into an imino group, trapping rhodamine 123 within mitochondria [118]. DHRH123 is oxidized by $H_2O_2$ in the presence of peroxidases, but it can also be oxidized by other reactive oxidants, as ONOO, $Fe^{2+}$, $Fe^{3+}$ in the presence of ascorbate or EDTA, cytochrome c, or HOCl [62,81]. DHRH123 is not directly oxidized by $H_2O_2$ alone, $O_2^{\bullet-}$ nor by the xanthine/xanthine oxidase system [107].

DHRH123 was described initially as a fluorogenic substrate for $H_2O_2$ [51] but it is now the most frequently used probe for measuring ONOO [118–120]. However, the oxidation of DHRH123 by ONOO is actually mediated by intermediate oxidants formed from the rapid and spontaneous decomposition of ONOO, and not induced directly by this radical [118–120].

### 3.2.4. Mitochondria Peroxy Yellow 1 and Related Arylboronate Fluorescent Probes

Recently, new $H_2O_2$ chemoselective probes have been developed based on the selective $H_2O_2$-mediated reaction of arylboronates into phenols [121,122]. Arylboronates are linked to fluorogenic moieties, so that reaction with $H_2O_2$ generates a fluorescent probe [121]. Arylboronate-based fluorescent reagents include peroxyfluor-2 (PF2), Peroxy Yellow 1 (PY1), Peroxy Orange 1 (PO1), Peroxyfluor-6 acetoxymethyl ester (PF6-AM) and Mitochondria Peroxy Yellow 1 (Mito-PY1) [121–126]. Adding appropriate functional groups results in organelle-specific targeting for measuring $H_2O_2$ with spatial resolution. Thus, Mito-PY1 and SHP-Mito [124–126] target mitochondria. while Nuclear Peroxy Emerald (NucPE) is suitable for nuclear targeting [121]. In addition, Ratio Peroxyfluor 1 (RPF-1) exhibits a ratiometric change in two fluorescent signals upon reaction with $H_2O_2$, allowing correlating fluorescence ratio to probe concentration [125].

### 3.2.5. Hydroethidine and MitoSOX Mitochondrial $O_2^{\bullet-}$ Indicators

Hydroethidine (HE), also know as dihydroethidium, is widely used for detecting $O_2^{\bullet-}$ anion [117,127,128]. HE is cell-permeant, and intracellular HE exhibits blue fluorescence, but once oxidized by $O_2^{\bullet-}$, it originates fluorescent 2-hydroxy-ethidium (E+), a ($\lambda$ excitation = 520 nm; $\lambda$ emission = 610 nm). E+ is retained in the nucleus by intercalating with DNA, a fact that enhances its fluorescence emission [107].

HE has been applied in studies of oxidative burst in leukocytes [128,129] and during inflammation [130–133]. HE has been used also for mitochondrial $O_2^{\bullet-}$ detection [117,134,135] although MitoSOX Red mitochondrial $O_2^{\bullet-}$ indicator provides more specific mitochondrial localization, as discussed later [117,136]. HE (and also MitoSOX Red) have been also used to detect mitochondrial $O_2^{\bullet-}$ generation associated to the induction and execution of apoptosis [134,135]. However, it has been shown that cytochrome c is able to oxidize HE, an aspect that might be relevant in conditions of apoptosis, where cytochrome c is released to cytosol [133]. Furthermore, HE can also be oxidized by several reactive species, including ONOO. Thus, HE should be considered mostly as an indicator of RONS generation [51,107,108,133].

MitoSOX Red mitochondrial $O_2^{\bullet-}$ indicator (MitoSOX Red) a cationic derivative of HE, has been used for detection of $O_2^{\bullet-}$ in live cells mitochondria [51,117,137]. MitoSOX Red contains a triphenylphosphonium cationic substituent targetting the probe to mitochondria, as a function of mitochondrial membrane potential [117]. Oxidation of MitoSOX Red by $O_2^{\bullet-}$ induces hydroxylation of the ethidium backbone at the 2-position, to yield a 2-hydroxyethidium substituent whose fluorescence spectral properties ($\lambda$ excitation = 488 nm; $\lambda$ emission = 610 nm) are similar to those of HE. In addition, MitoSOX Red shows also an absorption peak at 396nm, which may be used for more accurate detection of $O_2^{\bullet-}$ [117]. MitoSOX Red has been used for detection of mitochondrial $O_2^{\bullet-}$ production in a wide variety of cell types and conditions [107,117,136], including hypoxia [138]. Recently, the green-emitting version of MitoSOX Red, the probe MitoSOX Green ($\lambda$ excitation = 488 nm; $\lambda$ emission = 510 nm) has been marketed for detection of mitochondrial $O_2^{\bullet-}$ [117]. The probe reactivity towards $O_2^{\bullet-}$ is similar for both MitoSOX dyes and HE and, thus, the limitations described for HE apply also to MitoSOX Red and Green dyes [51,139].

Figure 6 presents the chemical structures of relevant fluorescent probes used to monitor ROS.

### 3.2.6. CellROX^TM Reagents

CellROX^TM includes a series of proprietary cell-permeant probes, that are weakly fluorescent in reduced state and exhibit photostable fluorescence upon oxidation by ROS [140–143]. CellROX Green becomes fluorescent ($\lambda$ excitation = 485 nm; $\lambda$ emission = 520 nm) only after binding to DNA, limiting its presence to the nucleus or mitochondria. This probe resists formaldehyde fixation and detergent treatment, allowing it to be it multiplexed with other compatible dyes and antibodies in preserved samples. CellROX Orange ($\lambda$ excitation = 545 nm; $\lambda$ emission = 565 nm) and CellROX Deep Red ($\lambda$ excitation = 640 nm; $\lambda$ emission = 665 nm) do not require binding to DNA for fluorescence emission and are localized in the cytoplasm [144].

### 3.2.7. ROS-ID^TM Reagents

The ROS-ID Total ROS/Superoxide detection kit is a proprietary system for real-time measurement of global ROS levels and specifically $O_2^{\bullet-}$ in living cells [145]. Oxidative Stress Green is a cell-permeable fluorogenic substrate that reacts directly with a wide range of RONS, generating a green fluorescent product [146]. The reagent Superoxide Orange is a cell permeable probe claimed to specifically react with $O_2^{\bullet-}$ generating an orange fluorescent product [147].

**Figure 6.** Chemical structures of relevant fluorescent probes used to monitor reactive oxygen species. For preparing this figure, we have used MolView, an open source chemical modeling package (http://molview.org/, accessed on 20 May 2023). The chemical structures have been retrieved directly via MolView from the PubChem Compounds database or sketched by us with MolView from the structures published in references [107–139]. The structures are identified by their chemical name or their tradename, with the abbreviation in parenthesis.

### 3.3. Detection of More Stable Products of ROS Reaction

3.3.1. Detection of Lipid Peroxidation

Peroxyl radicals are the end-products of the decomposition of various peroxides and hydroperoxides, including lipid hydroperoxides. The hydroperoxyl radical can be also considered as the protonated form of $O_2^{\bullet-}$, accounting for near 0.3% of the $O_2^{\bullet-}$ present in the cytosol [117].

*cis*-Parinaric Acid

*cis*-Parinaric acid is a naturally occurring fluorescent 18-carbon polyunsaturated fatty acid, with conjugated double bonds in positions 9, 11, 13 and 15 [117]. The unusual *cis-trans* structure of the conjugated tetraene generates a fluorophore with natural fluorescence (λ excitation = 320 nm; λ emission = 432 nm) that is lost upon oxidation. Thus, the fluorescent and peroxidative properties of *cis*-parinaric acid are intrinsic of the conjugated system of double bonds. Exogenous *cis*-parinaric targets to phospholipids of cell membranes, where its conformation and mobility are comparable to those of endogenous phospholipids [117].

*cis*-Parinaric has been repeatedly used for the measurement of phospholipase activity, lipase activity, and lipid peroxidation in different cell systems and conditions [148–150]. However, there are some limitations associated with the use of this probe in FCM, such as its excitation by UV lasers, still absent in many flow cytometers. In addition, *cis*-parinaric is very sensitive to air and undergoes photodimerization under illumination, resulting in spontaneous loss of fluorescence and overestimation of lipid peroxidation [117].

BODIPY 581/591 C11 and Related Probes

4,4-Difluoro-5-(4-phenyl-1,3-butadienyl)-4-bora-3a,4a-diaza-s-indacene-3-undecanoic acid (BODIPY 581/591 C11) is a fluorescent probe (λ excitation = 510 nm; λ emission = 595 nm) used for assessing lipid peroxidation and antioxidant efficiency [117,150]. BODIPY 581/591

C11 contains a non-polar, long-chain (C11) unsaturated tail, which targets the probe to the lipophilic domain of the membranes, while the conjugated double bonds in the fluorophore(BODIPY) make it susceptible to oxidation [107,117]. Upon oxidation, BODIPY 581/591 C11 undergoes a shift fluorescence emission from red to green, thus enabling a more accurate ratiometric analysis of ROS-induced probe oxidation [117].

BODIPY 581/591 C11 is oxidized by peroxyl, OH$^{\bullet}$ radicals and ONOO, but it is insensitive to $H_2O_2$, $^1O_2$, $O_2^{\bullet-}$, NO in the presence of transition metals, and to hydroperoxides in the absence of transition metals [151].

Lipid peroxidation has been detected in cell membranes using BODIPY 581/591 C11 [152–154] and other similar BODIPY derivatives, such as BODIPY493/503 [155], BODIPY FL EDA [117], a water-soluble BODIPY dye, or BODIPY FL hexadecanoic acid [117].

Lipophilic Fluorescein Derivatives

The probe 5-(N-dodecanoyl) aminofluorescein (C11-Fluor) is a lipophilic fluorescein-derivative that remains associated stably and irreversibly to cell membranes. C11-Fluor has been used in FCM for determining membrane-lipid peroxidation [43,156]. Other lipophilic derivatives of fluorescein include 5-hexadecanoylaminofluorescein (C16-Fluor), 5-octadecanoyl-aminofluorescein (C18-Fluor) and di-hexadecanoyl-glycerophosphoethanolamine (Fluor-DHPE) [157].

Figure 7 summarizes the chemical structures of relevant fluorescent probes used to detect lipid peroxidation.

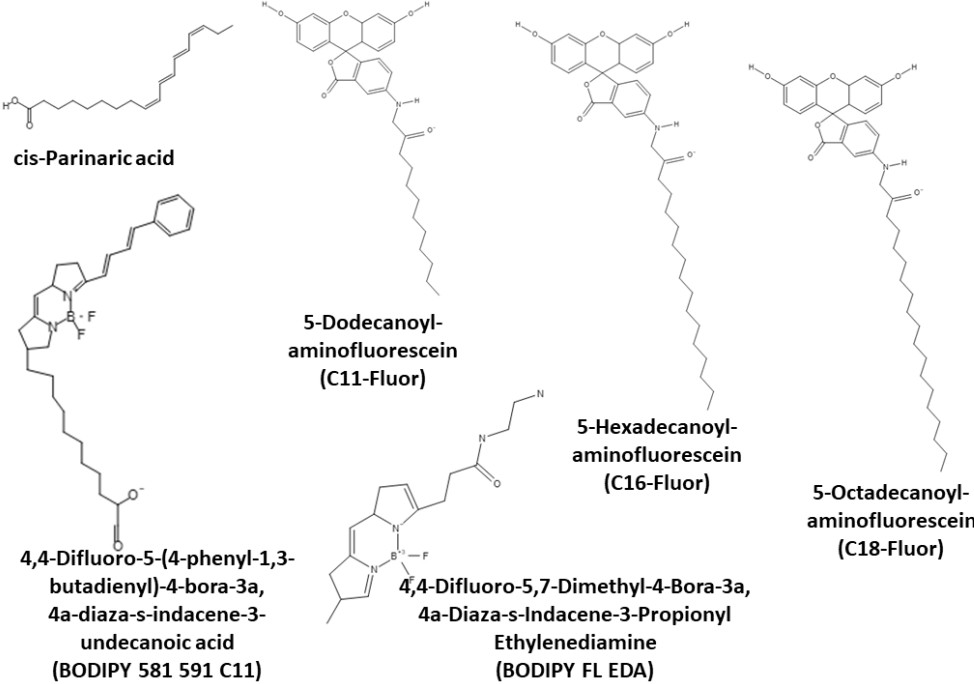

**Figure 7.** Chemical structures of relevant fluorescent probes used to detect lipid peroxidation. For preparing this figure, we have used MolView, an open source chemical modeling package (http://molview.org/, accessed on 20 May 2023). The chemical structures have been retrieved directly via MolView from the PubChem Compounds database or sketched by us with MolView from the structures published in references [148–157]. The structures are identified by their chemical name or their tradename, with the abbreviation in parenthesis.

3.3.2. Detection of Metabolic Derivatives of Peroxidized Lipids
Detection of 4-Hydroxy-2-nonenal by Immunofluorescence

Several aldehydes are formed as end-products of the peroxidation process. 4-hydroxy-2-nonenal (4-HNE) is an unsaturated aldehyde arising from peroxidation of ω-6 unsat-

urated fatty acids. 4-HNE is highly reactive towards free SH- groups of proteins and is highly cytotoxic and genotoxic [158,159].

Several monoclonal antibodies recognize the adducts formed by reaction of 4-HNE with histidine, lysine, and cysteine in proteins are commercially available [160]. These antibodies have been conjugated with different fluorochromes and can be used with high specificity for detecting the cellular consequences of lipid peroxidation [161].

Immunofluorescent Detection of Oxidized Bases in DNA

8-oxodeoxyguanine (8-oxoDG) is an oxidized DNA base derived from the attack of OH$^\bullet$ radical to the C-8 position of guanine, resulting in a C8-OH-adduct radical. Thus, 8-oxoDG is a major form of oxidative DNA damage by free radicals and is considered a sensitive and specific indicator of DNA oxidation [162,163].

The OxyDNA assay is based on the specific binding of a FITC-conjugated monoclonal antibody to 8-oxoDG adducts in the DNA of fixed and permeabilized cells [164]. This assay has been used to detect oxidative genotoxicity in vitro [165], including environmental studies [166]. Of particular interest, the OxyDNA assay has been used in a number of fertility studies aimed to detect oxidative stress during sperm cryopreservation [167] and to assess the relation of oxidative DNA damage to fertility in humans [168,169] and animals [170].

*3.4. Assessment of Antioxidant Defenses: Glutathione and Thiols*

Thiol (SH) groups, and especially glutathione (GSH), are nucleophilic species that protect against toxicity, mutagenicity, or transformation by ionizing radiation and many carcinogens [40]. The availability of many SH-reactive fluorescent probes since the early 1980s has provided FCM assays for GSH [171,172] and free SH- groups [173] in living cells. Additionally, analysis of intracellular levels of GSH and of GSH S-transferase activity (GST) are now relevant applications of FCM in oxidative stress and drug resistance [174]. Cytometric assays for GSH and intracellular SH-groups have been reviewed previously+ [175–178].

The cell-permeant bimanes, including monobromo-bimane (mBrB) and the more selective monochloro-bimane (mClB), are the most used probes for cytometric analysis of GSH and GST. Both UV-excited probes are essentially nonfluorescent until conjugated to GSH [174–177]. o-Phthaldialdehyde, another UV-excited probe, reacts with both the SH- and the amine functions of GSH, yielding a cyclic derivative with excitation and emission maxima shifted from those of the protein adducts, improving the specificity of GSH detection [174–177]. ThiolTracker Violet (λ excitation = 405 nm; λ emission = 525 nm) is up to 10-fold brighter than the bimanes. This cell-permeant probe resists formaldehyde fixation and detergent extraction, allowing the analysis of fixed cells [174,178].

GSH has been determined also with visible light-excitable probes, including Mercury Orange [173], MitoTracker Green FM, 5-chloromethylfluorescein diacetate (CellTracker Green CMFDA), and chloromethyl SNARF-1 acetate. These probes form adducts with SH groups of intracellular proteins and are well retained by viable cells. CellTracker Green CMFDA is brighter than mClB and is much more specific for GSH than for protein SH-groups [174]. The GSH adduct of chloromethyl SNARF-1 is fluorescent beyond 630 nm, allowing multicolor protocols and reducing the impact of cell autofluorescence.

Figure 8 shows the chemical structures of relevant fluorescent probes used to assess GSH and SH groups.

**Figure 8.** Chemical structures of relevant fluorescent probes used to assess antioxidant defenses. For preparing this figure, we have used MolView, an open source chemical modeling package (http://molview.org/, accessed on 20 May 2023). The chemical structures have been retrieved directly via MolView from the PubChem Compounds database or sketched by us with MolView from the structures published in references [171–178]. The structures are identified by their chemical name or their tradename, with the abbreviation in parenthesis.

## 4. FCM in Oxidative Stress Research

As commented above, FCM analysis of oxygen, ROS and oxidative stress is a complicated issue, due to the low concentration, short half-life and multiple interactions among ROS and RNS, as well as by limitations intrinsic to the fluorescent probes or resulting from the experimental conditions. Such limitations are potential sources of artifacts that challenge accurate analysis of intracellular ROS and require a careful experimental design and a cautious interpretation of the results [53].

### 4.1. Variability in Half-Life and Intracellular Sources of ROS

While most ROS are short-lived molecules, there is large heterogeneity in their lifetime and the intracellular compartments where they are produced and consumed [51,108]. ROS of low reactivity may accumulate in the cells, but the more reactive ROS will reach a steady state, with a rate of their generation equal to the rate of disappearance by self-reaction, by reaction with cellular components or by interaction with the exogenous probes.

Specific intracellular locations of ROS and RNS can be determined by appropriate molecular design of the fluorogenic substrates [51–53]. Introduction of chemical modifications in their molecular scaffolds renders the probes permeable to plasma- and organelle membranes, thus allowing to target them to specific intracellular environments [51,52,107,108]. For instance, dihydrocalcein accumulates in mitochondria, in contrast to $H_2DCF$-DA which usually localizes in the cytoplasm [179]. However, preferential localization of $H_2DCF$-DA in the mitochondria of rat cardiomyocytes has been reported [180].

To prevent or minimize these complications, FCM techniques based in real-time measurements (Figure 3) [181] and imaging cytometry of intracellular location of ROS may be used [50,53].

*4.2. Interactions among and between ROS and RNS*

Mitochondrial respiration exemplifies ROS interplay, as $O_2^{\bullet-}$ anion, $H_2O_2$ and $OH^\bullet$ radical are produced in sequence by a series of partial reductions of the $O_2$ molecule. Incorporation of an electron into $O_2$ generates to $O_2^{\bullet-}$ anion, which is a poorly reactive radical but which can oxidize SH groups and ascorbic acid [5,10,40]. $O_2^{\bullet-}$ gives rise to $H_2O_2$ either by spontaneous reaction or by the action of superoxide dismutase. $H_2O_2$, in turn, reacts with different molecules to produce peroxyl radicals that will eventually release $OH^\bullet$ radicals during their metabolism. Moreover, by way of the Fenton reaction, $OH^\bullet$ radicals are produced when $H_2O_2$ and a transition metal, such as $Fe^{2+}$, react together, yielding $Fe^{3+}$ that consumes $O_2^{\bullet-}$ for recycling $Fe^{2+}$. In the Haber–Weiss reaction, $O_2^{\bullet-}$ and $H_2O_2$ react to produce $OH^\bullet$ radicals [5,10].

On the other hand, the interaction of ROS with nitrogen derivatives can generate RNS. For instance, many cell types produce NO from L-arginine by different isoforms of NO synthetase. NO is a weak reductor and reacts with $O_2$ to form $NO_2$, but reacts much faster with $O_2^{\bullet-}$ to produce ONOO, a strong oxidant [181].

*4.3. Influence of the Probes on the Experimental System*

Many reagents used for FCM analysis of ROS are chemically reduced fluorogenic substrates, intrinsically susceptible to auto-oxidation, which usually produces $^1O_2$, $O_2^{\bullet-}$ and $H_2O_2$. If the rate of auto-oxidation is relevant, it may result in artifactual detection of ROS and a higher fluorescent background, a problem especially evident for many commonly used probes, such as HE or MitoSOX Red [51,107,108].

The concentration of the probe may affect the stoichiometry of the process under study. For instance, the stoichiometry of the reaction between HE and $O_2^{\bullet-}$ depends on the ratio of $O_2^{\bullet-}$ flux and HE concentration. Due to HE-catalyzed $O_2^{\bullet-}$ dismutation, the efficiency of HE oxidation decreases at higher rates of $O_2^{\bullet-}$ generation and high HE concentrations might lead to increases in fluorescence emission independent of the $O_2^{\bullet-}$ concentration [107].

Fluorescent probes at high concentration, may interfere with cell functions and be toxic [47,53]. For example, $H_2DCF$-DA undergoes auto-oxidation and photosensitizes cells when irradiated with UVA [108]. In addition, probes may affect the activity of ROS-producing enzymes. For instance, $H_2DCF$-DA can be a source of electrons for the oxidation of arachidonic acid by prostaglandin H synthase [108], and dihydrocalcein has been reported to inhibit mitochondrial-chain complex I [51].

Another aspect to take into account in the use of probes with mitochondrial localization, such as MitoPY1, DHRH123 or MitoSox Red is the possible effect of such probes on the mitochondrial membrane potential. Mitochondrion-targeted chemical probes are molecules with cationic groups (triphenylphosphonium for MitoSox Red and MitoPY1; imino group for DHRH123) that are attracted to the negative electric charge generated by mitochondrial respiration. An excess of probe input of positive charge could reduce the net negative charge of the mitochondria and decrease the uptake of new cationic molecules.

*4.4. Cell Integrity and Intracellular Retention of Probes*

Loss of intracellular fluorogenic substrates or fluorescent probes from injured cells is a common problem in FCM analysis, most specially in pharmaco-toxicological studies. Passive leakage is always present in necrotic- or apoptotic cells, due to increased plasma membrane permeability [47,48,53]. In oxidative-stress related studies, therefore, a certain degree of damage to plasma membrane should be expected, with leakage of intracellular probes or their oxidation products leading to artifacts or erroneous interpretation of results.

On the other hand, the activity of multidrug transporters in the plasma membrane of many cell types results in probe extrusion, and leads to underestimate oxidative stress [151], since multidrug-resistant cells would appear to contain less ROS. Fluorescent molecules such as rhodamine 123 and ethidium are good substrates for P glycoprotein, while fluorescein and dihydrofluorescein are substrates for MRP1 [182]. Dihydrocalcein has been

preferred to $H_2$DCF-DA because calcein, its oxidation product, is believed to not leak out of cells. However, calcein is also a good substrate for MRP1 and MRP2 transporters [43].

### 4.5. Experimental Artifacts

The photochemical reaction of diverse components of culture media may spontaneously generate ROS [43,183]. Xenobiotics and intracellular compounds such as catechols, dopamine, hydralazine and molecules with SH- groups may also produce significant ROS upon interaction with the culture media [43]. Interestingly, the presence of ROS has been detected even in natural environments, such as seawater [184,185].

On the other hand, binding to macromolecules of fluorogenic substrates and fluorescent probes may result in quenching of the probe fluorescence. For example, quenching of $H_2$DCF-DA fluorescence has been reported after by binding to native- or glyoxal-modified human serum albumin [43,186].

### 4.6. Intrinsic Limitations of Fluorogenic Substrates and Probes

4.6.1. Probes Used for Detection of $H_2O_2$ and Organic Peroxides

$H_2$DCF-DA is amply used for detecting intracellular $H_2O_2$ and oxidative stress. Traditionally, $H_2$DCF-DA and DHRH123 are assumed to be oxidized by $H_2O_2$ and organic peroxides [109,117]. However, these probes do not react with $H_2O_2$ in the absence of peroxidases [51,108]. While $H_2$DCF-DA oxidation also occurs by action of $H_2O_2$ or $O_2$ in the presence of $Fe^{2+}$, the $OH^\bullet$ radical is the actual ROS responsible for such oxidation [107].

Since the oxidation of $H_2$DCF-DA and DHRH123 by $H_2O_2$ under physiological conditions requires peroxidase-dependent systems, enzyme activity may become a limiting factor and thus the rate of probe oxidation might be rather considered as a measure of the peroxidase activity. Moreover, $H_2$DCF-DA and DHRH123 can be oxidized not only by the peroxidases, but by other related enzymes, such as xanthine oxidase, superoxide dismutase and cytochrome c [107].

$H_2$DCF-DA and DHRH123 are not oxidized to a significant extent by NO or $O_2^{\bullet-}$, but they are oxidized very efficiently by ONOO via radicals generated during ONOO decomposition [119,120]. $H_2$DCF-DA undergo photoreduction by visible light or UVA radiation [187]. This mechanism may generate a semiquinone radical from DCF, which in turn originates $O_2^{\bullet-}$ by reaction with $O_2$. Sequentially, the dismutation of $O_2^{\bullet-}$ generates $H_2O_2$, which leads to an artificial increase in $H_2$DCF oxidation and to a cycle of amplification of DCF fluorescence.

Mito-PY1 and other boronate-derived probes used for the analysis of intramitochondrial generation of $H_2O_2$ [121–123] also react with ONOO a million times faster than they do with $H_2O_2$ [188]. Because of this reactivity, when using boronate-based fluorescent probes it is very important to include additional controls, such as inducing expression of catalase or using a ONOO sensitive probe.

4.6.2. Probes Used for Detection of $O_2^{\bullet-}$

HE and MitoSOX Red are widely used for detection of intracellular- and mitochondrial $O_2^{\bullet-}$ [51,108]. Ethidium (E+), the red fluorescence product of the oxidation of HE, is usually considered the proof of intracellular $O_2^{\bullet-}$ formation. However, E+ is not formed from the direct oxidation of HE by $O_2^{\bullet-}$ [189,190]. Instead, 2-hydroxyethidium (2-OH-E+), a different product with similar fluorescence characteristics, is formed by reaction of HE with $O_2^{\bullet-}$ [190]. E+ and other dimeric products, but not 2-OH-E+, are generated by the reaction between HE and oxidants such as ONOO, $OH^\bullet$, $H_2O_2$, and peroxidase intermediates. Thus, 2-OH-E+ should be considered only a qualitative indicator of intracellular $O_2^{\bullet-}$ [51,191].

HE is oxidized directly by heme proteins, including ferricytochrome c [127]. Oxidation of the probe by cytochromes c, c1, b562, b566 and aa3 is oxygen-independent while oxidation by met-hemoglobin and met-myoglobin is strictly oxygen-dependent, with products consisting of a mixture of molecules resulting from 1 to 4-electron abstraction from HE.

Although they are different from the $O_2^{\bullet-}$ oxidation product, their excitation/emission peaks are close to those generated by $O_2^{\bullet-}$ [51,53].

Because of its positive charges, MitoSOX Red reacts slightly faster with $O_2^{\bullet-}$ compared to HE [101]. MitoSOX Red reacts specifically with $O_2^{\bullet-}$ and forms a red fluorescent product, 2-hydroxy-mitoethidium (2-OH-Mito-E+), with fluorescence spectral properties that overlap with those of Mito-E+, the nonspecific product of MitoSOX Red. Thus, the red fluorescence formed from Mito-SOX localized in mitochondria is not an accurate indicator of mitochondrial formation of $O_2^{\bullet-}$, as it might arise also from oxidation of Mito-SOX induced by one-electron oxidants (such as cytochrome c, peroxidase, and $H_2O_2$) [51,137,189–192].

### 4.6.3. Probes Used for Detection of Lipid Peroxides

The presence of four double bonds in *cis*-parinaric acid makes this probe very susceptible to oxidation if not carefully protected from air [43,117], and *cis*-parinaric preparations should be handled under inert gas and solutions prepared with degassed buffers and solvents. In addition, *cis*-Parinaric acid is photolabile and undergoes photodimerization under intense illumination, resulting in loss of fluorescence [117].

BODIPY581/591 C11 is photosensitive and degrades under intense illumination [117]. In addition, BODIPY581/591 C11 is more sensitive to oxidation than endogenous lipids and its use may overestimate oxidative damage and underestimate antioxidant protection [117].

### 4.6.4. Probes Used for the Determination of GSH

The fluorescent reagents designed to measure GSH may react with other free or protein-bound intracellular SH-groups [173,176,193]. An important aspect in the use of GSH reagents is the large variability of cellular GSH content and the expression of GST isozymes among different species and tissues [193]. For instance, mClB is highly selective for GSH in rodents, but has a low affinity for human GST [176].

## 5. Recommendations for Performing FCM Analysis of ROS, RNS and Oxidative Stress

According to Woolley et al. [194], a fluorescent probe must fulfill a series of criteria in order to be an "ideal" indicator of ROS. These criteria include probe selectivity for a particular species of ROS, fast and reversible kinetics when reacting with ROS, and adequate subcellular compartmentalization. The probe should be excitable at a visible wavelength, to be resistant and not to show any toxicity in general and phototoxicity in particular. To the above criteria, it must be added that of the probe not interfering with the biological process under study [47,48].

Following these principles, we have recently performed a systematical study to detect experimental issues related to the specificity of fluorescent probes and the involvement of different ROS in *Escherichia coli* [126] and eukaryotic [195] models of oxidative stress. Our results, as summarized below, may provide recommendations for proper design of cytometric studies of oxidative stress to prevent or minimize experimental errors.

### 5.1. Inclusion of Experimental Controls

Resulting from the limitations and caveats presented above, the inclusion of suitable positive and negative controls is essential when performing FCM basic experiments or clinical determinations related to ROS and oxidative stress. When possible, direct visualization of intracellular generation of ROS by co-localization techniques is highly recommendable [50,53]. A detailed discussion of possible controls in such studies is beyond the scope of this review, as the biochemical complexity of experimental oxidants and antioxidants is comparable to that of their biological counterparts [40,196,197].

#### 5.1.1. Positive Controls

The most frequent controls are positive controls, i.e., those molecules or complex systems that increase the intracellular level of ROS or mimic the cellular effects of oxidative

stress. Pro-oxidants are chemicals that induce oxidative stress, either by generating ROS or by inhibiting antioxidant systems [40,196,197].

To mimic mitochondrial $H_2O_2$ production, cells can be treated with rotenone, an inhibitor of the complex I in the respiratory chain [123]. Peroxyl radicals, including alkylperoxyl and hydroperoxyl radicals can be generated from exogenous compounds such as 2,2′-azobis(2-amidinopropane) and from hydroperoxides such as tert-butyl hydroperoxide(t-BOOH) or cumene hydroperoxide (CHP) [123,126]. The $OH^{\bullet}$ radical can be generated from $O_2^{\bullet-}$ donors (e.g., plumbagin or menadione) [123,181] or by exogenous $H_2O_2$ via a Fenton reaction catalyzed by $Fe^{2+}$ or other transition metals, as well as by ionizing radiation [123,126]. $O_2^{\bullet-}$ can be effectively generated by the hypoxanthine–xanthine oxidase system [198].

Many xenobiotics, including anticancer agents such as anthracyclines and *cis*-platin [199], and natural redox active toxins, such as pyocyanin, [200] generate ROS and are used as positive controls.

When using prooxidant molecules with different physico-chemical properties, as $H_2O_2$ and organic peroxides, such as t-BOOH and CHP, it is important to consider their lipophilicity and their ability to penetrate or being transported across cell membranes. Thus, at comparable oxidant concentrations, the oxidative stress induced in membrane is always much higher for t-BOOH and CHP, while intracellular oxidative effects are better detected with $H_2O_2$ [3,126]. Moreover, the intracellular action of t-BOOH and CHP is gradual, generating ROS that produce membrane-initiated oxidative stress, which increases over time to a maximum and then decreases. On the contrary, $H_2O_2$ reacts rapidly by generating ROS that are highly reactive towards intracellular components. $H_2O_2$-induced oxidative stress usually is maximal immediately after addition and rapidly disappears.

Intracellular levels of ROS can be also increased by decreasing or inhibiting antioxidant defense. A convenient strategy involves depletion of intracellular GSH stores by inhibiting GSH biosynthesis or by accelerating GSH oxidation [201]. Additionally, inhibitors of antioxidant enzymes, such as superoxide dismutase [202] and catalase [203] have been used to increase intracellular ROS and induce oxidative stress.

5.1.2. Negative Controls

Negative controls are included in order to reduce the levels of ROS or to atenuate their biological effects. If possible, controls should be informative about the particular reactive species or enzyme system are involved, but in most cases, controls do not reachthat degree of specificity [51,52,108].

Antioxidants can be categorized as enzymatic and nonenzymatic [196]. Enzymatic antioxidants work by transforming oxidative products to $H_2O_2$ and then to $H_2O$, in a sequential process. Cell-permeable forms of antioxidant enzymes, such as polyethylenglycol-superoxide dismutase [204] can be also used to decrease specifically intracellular ROS.

Non-enzymatic antioxidants work by disrupting free-radical initiated chain reactions. Such antioxidants may be hydrophilic (e.g., ascorbic acid, N-acetyl cysteine, GSH-esters) or lipophilic (e.g., $\alpha$-tocopherol and Trolox) [40]. In general, hydrophilic antioxidants react with oxidants in the cytosol, while lipophilic antioxidants protect membrane compartments from lipid peroxidation [40]. The chelators of transition metals exert also antioxidant effects, based upon the attenuation of Fenton-type reactions [40].

Regarding chemical antioxidants as negative controls, it should be kept in mind that reducing agents may become pro-oxidants. For instance, ascorbate has antioxidant activity when it reduces oxidants such as $H_2O_2$, but it can also reduce metal ions, generating free radicals by the Fenton reaction [40].

As for the specificity of antioxidants, all organic compounds react with $OH^{\bullet}$ radicals with rate constants approaching the diffusion limit. Thus, when in aqueous solution, no compound would have more $OH^{\bullet}$ radical scavenging activity than other molecules already present in the biological system [40]. On the contrary, $\alpha$-tocopherol, due to its specific uptake into membranes, and its fast reaction kinetics with lipid hydroperoxyl

radicals (compared with their propagation reaction) is an effective chain breaker of lipid peroxidation [40].

5.1.3. Genetically-Modified Organisms as Controls

Genetically-modified organisms are alternative approaches to chemical or biochemical positive and negative controls in oxidative stress studies. For instance, Guo et al. [124] generated cytoplasmic $H_2O_2$ in primary astrocytes. by transducing cytoplasmic D-Amino acid oxidase (DAAO). DAAO oxidatively deaminates d-amino acids using FAD as electron acceptor. At the same time, DAAO uses $O_2$ to oxidize FAD, thus generating $H_2O_2$ in a dose-dependent manner.

Bacterial strains, mostly *E. coli*, with genetical modifications in genes related to the antioxidant defense have been used repeatedly as models for studying ROS mechanisms and oxidative stress [70,126,205,206]. However, functional FCM assays in live bacteria are limited by the impaired penetration of vital dyes across the cell wall, thus demanding permeabilization procedures that may affect cell physiology, leading to aggregation or cell lysis [206].

*E. coli* B is for the biological basis for mutagenic assays [207–210], while *E. coli* K12 strain is applied mostly to genetic and biochemical studies. *E. coli* B WP2 strains express constitutively different cell-wall and outer-membrane lipopolysaccharide components that result in increased membrane permeability, thus making this strain more suitable for both protein secretion and uptake of exogenous chemicals, as recently confirmed by multi-omic analysis comparing *E. coli* B and *E. coli* K12 strains [211].

In accordance with these findings, we have demonstrated the utility of *E. coli* B as a naturally-permeable strain in straight-forward functional FCM assays [206]. The *E. coli* B strain IC188 is stained more efficiently with vital fluorochromes than the *E. coli* K-12 strain AB1157, while maintaining similar membrane potential. In addition, IC188 strain is more sensitive than AB1157 to exogenous oxidants, showing its suitability as a biosensor of oxidative stress [205]. On this background, we developed a series of of *E. coli* B WP2 strains [209] based on the inactivation of the oxyR operon, a main sensor of oxidative stress [212]. When studied by FCM, such oxyR-deficient strains show enhanced sensitivity to oxidative stress and increased accumulation of intracellular ROS [206]. More recently [126], we have characterized two novel biosensors derived from the *E. coli* B parental wild type (strain IC188) deficient either in the OxyR function (strain IC203) or simultaneously in OxyR, SodA and SodB functions (strain IC5233). On these models, we have quantified the intracellular levels of ROS by FCM using ROS-sensitive fluorescent probes after exposure to relevant xenobiotics differing in solubility and prooxidant mechanisms.

*5.2. Choice of Fluorescent Probes*

Our results have shown a higher sensitivity and specificity of the green-emitting Mito-PY1 probe for the detection of $H_2O_2$, both in the bacterial model [126] and in RT-FCM experiments with hepatoma cells [195]. On the contrary, our results showed that Mito-PY1 is not be the fluorescent probe of choice for assessing oxidative stress induced by organic peroxides or redox cycle compounds. In fact, Mito-PY1 is a chemoselective fluorescent reporter of the arylboronate family, with improved selectivity for $H_2O_2$ over other ROS [121]. In addition, Mito-PY1 is a probe designed for its mitochondrial localization, which is why it probably detects $H_2O_2$ present in the mitochondria [123], which would explain the low response of Mito-PY1 to exogenous organic hydroperoxides and redox cycle compounds, which initially generate $H_2O_2$ in membrane environments and initiate chain reactions that lead to the diffusion of ROS and the generation of the highly reactive $OH^\bullet$ radical [126].

Our results with *E. coli* [126] and cancer cell lines [195] also showed that the green-emitting probes $H_2DCF$-DA and DHRH123 are more sensitive to organic peroxides, including t-BOOH and CHP, than to $H_2O_2$, confirming their lack of selectivity of $H_2DCF$-DA for $H_2O_2$. These results confirm the caveats regarding $H_2DCF$-DA and DHRH123 [213], possi-

bly the most widely used probes to detect intracellular oxidative stress [53]. Accordingly, both probes should be used rather than Mito-PY1 for studies involving organic peroxides that generate oxidative stress through $H_2O_2$-indepent mechanisms.

The orange-red emission fluorochromes HE and its mitochondrial localization derivative MitoSOX Red are widely used in FCM as indicators of high specificity towards $O_2^{\bullet-}$ [127,137]. In line with this idea, a greater response has been shown in the MitoSox Red probe when inducing with menadione and greater HE specificity with plumbagin, both $O_2^{\bullet-}$ donor xenobiotics. Based on these observations, these probes could be recommended for use in oxidative stress studies in which it is desired to quantify $O_2^{\bullet-}$ generation. A study in HT22 cells exposed to menadione showed an accumulation of $O_2^{\bullet-}$ in mitochondria [214]. This is related to what was observed in Jurkat cells, with higher ratios of MitoSOX Red, a mitochondrial probe, after incubation with menadione.

Consistent with the specificity of HE and MitoSOX Red, the results with Jurkat and N13 cells [195] show the very low response of both probes to treatment with $H_2O_2$ but dose-dependent weak responses to all agents prooxidants used in the Jurkat cell model. On the other hand, in the bacterial strain model [126], although we do not have data for MitoSox Red, the HE responded to all the oxidizing agents used, with different levels of intensity, which casts doubts to the direct applicability of this probe as a specific sensor. of $O_2^{\bullet-}$ and requires a careful review of the chosen experimental model.

The results obtained in our systematic studies with the CellROX® and the ROS-ID® reagents families of products [126,196] have shown that, in general, all these probes present medium to low intensity in the responses to the different prooxidants tested, with a very broad specificity towards them, which is in accordance with the manufacturers' warnings, included in the documentation of the reagents and kits.

When fine-tuning multiparametric kinetic and endpoint assays of oxidative stress [196] we have found striking discrepancies in the responses of the ROS-sensitive probes analyzed individually or in spectrally compatible pairs. Since in such experiments fluorescence compensation was performed between the pairs of fluorochromes, the data obtained seemed to suggest the existence of biochemical interferences between the probes of each pair and/or of the probes with ROS metabolism and oxidative processes induced by xenobiotics [57]. Thus, while the different fluorescent probes responded individually to the addition of pro-oxidant compounds as expected, the probes in spectrally-compatible combination with other probes showed important increases or reductions in the fluorescence generation rate induced by the same prooxidant. Such observations are compatible with the consumption or generation of ROS by some of the probes in the tested pairs. However, the interpretation is complicated and requires further experiments, which may include more specific control systems for generating ROS as well as novel, single-cell-based imaging technologies, such as multi-spectral-imaging flow cytometry [50].

### 5.3. Titration of Fluorescent Probes

Titration of fluorescent probes is a very important step when setting up a florescence-based determination in FCM. Probes must be titrated properly in order to define their optimal staining concentration for avoiding saturation and suboptimal detection of the relevant biological parameters [58].

A key point to address in functional FCM experiments is to avoid, or at least, to assess, the influence of the probes on the experimental system. In the case of oxidative stress, all reduced fluorogenic substrates may be subject to auto-oxidation, which usually produces $^1O_2$, $O_2^{\bullet-}$ and by its dismutation, $H_2O_2$. If the auto-oxidation rate is significant, it may lead to artifactual detection of ROS and higher fluorescent background, a problem especially important for probes such as HE [46]. The concentration of the probe is also relevant, as it may affect the stoichiometry of the process under study. In fact, the probes themselves may affect the activity of ROS-producing enzymes [53]. Finally, fluorescent probes at high concentration may be toxic to the cells [47].

To minimize the impact of artifacts derived from excessive probe concentration, fluorochrome titration is an essential procedure [58]. By defining the minimal concentration of a given fluorescent probe required for sensitive detection of a given ROS, the non-specific detection of this ROS can be minimized, as demonstrated by the low level of intracellular fluorescence observed in cells not exposed to exogenous peroxides (Figure 4).

*5.4. Range-Finding Experiments and Exclusion of Dead/Injured Cells*

A significant number of oxidative stress studies by FCM involve detection of ROS generation by in vitro exposure to xenobiotics [48]. In the pharmaco-toxicological context, it is important to quantify the toxic potency of the drug [215] and still the most used method for quantifying viable cells is the classical colorimetric assay with 3-(4,5-dimethylthiazol-2-yl)-2,5-diphenyltetrazolium bromide (MTT) [216]. However, this is a bulk method that provides only an average estimation of the viable-cell number based on the dehydrogenase activity ability of metabolically competent cells. On the contrary, setting up cell function analysis by FCM requires clear identification of single live cells in order to exclude cell aggregates and dead or dying cells [55,58]. Moreover, identifying live cells by FCM provides an alternative method to assess the toxic potency of xenobiotics [48,217]. While the FCM procedures for single live-cell identification are relatively easy when analyzing human or mammalian cells [48,55,58], they become complicated for bacteria, mostly due to their smaller size and different permeability to viability dyes than in eukaryotic cells [126,218].

According to these aspects, performing a range-finding experiment for determining the toxicity (e.g., the IC50 value) of a test- or positive-control chemical is essential to establish toxic-exposure conditions that allow the presence of identifiable and selectable live cells in the sample (Figure 4). This will require only including viability markers that are spectrally compatible with the other fluorescent probes that define the experiment [48,55,58,126,218]. In this way, the loss of intracellular fluorogenic substrates or fluorescent probes due to plasma membrane lesion may be detected and minimized, as commented in previous sections of this review [47,48,53].

*5.5. Data Generation, Presentation, Data Analysis and Publication*

As for all the laboratory technologies, data generation in FCM is highly dependent on the quality issues that define the pre-analytical phase (i.e., sample obtention, storage, handling and preparation) and the analytical phase (i.e., instrument control and optimization) should be carefully kept [47,48,53,56,58]. In addition, the special way in which data are generated from specific cell subpopulations (i.e., single cells versus cell aggregates; live cells versus non-viable cells; cells with a particular phenotype or biological properties) requires that the procedures followed for such identification are clearly presented and justified in the post-analytical phase [47,48,58] in order to assess the reproducibility of the results and their interpretation. These considerations are at the basis of the publication policy of cytometry-specialized journals, which require that data suitable for independent reproduction of the results be made available prior to publication and recommend that authors submit their data files to a repository [219] prior to submission. In addition, such journals require submitting a file with Minimum Information about a flow cytometry experiment (MIFlowCyt Standard), that has been set by the International Society for the Advancement of Cytometry (ISAC). MIFlowCyt states the minimum information required to report FCM experiments to allow for future independent validation and interpretation of experimental data [220].

The issues related to the presentation of FCM data in the context of studies of ROS generation and oxidative stress are exemplified in Figure 9.

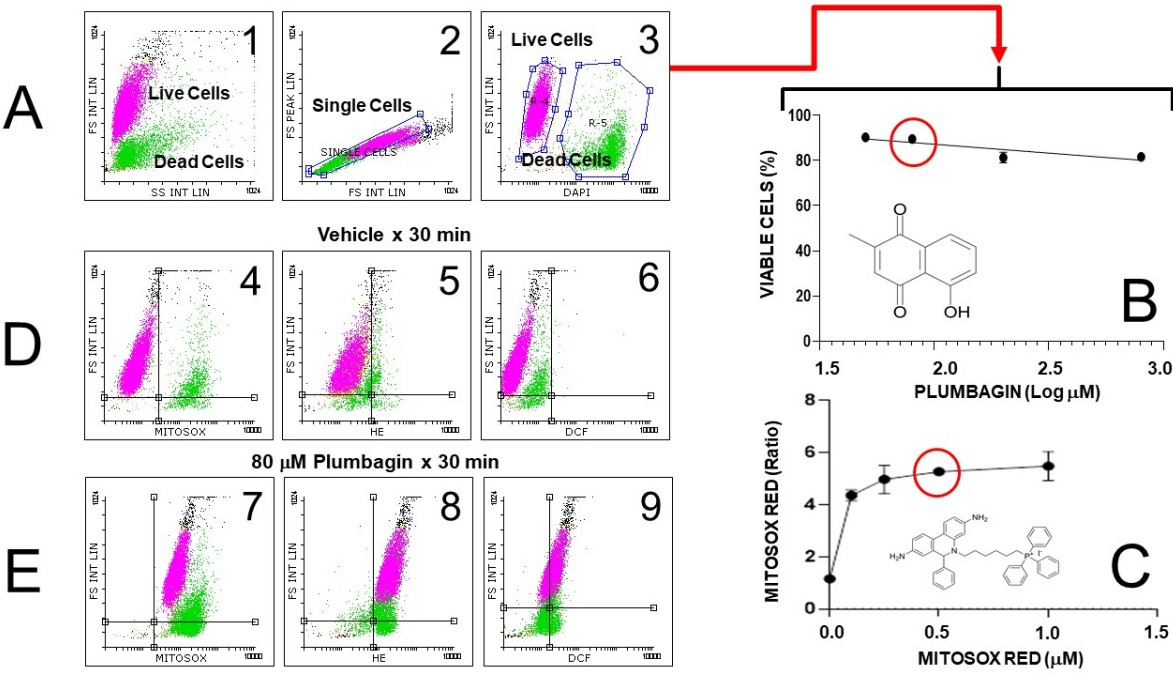

**Figure 9.** Scheme of the most relevant aspects of the experimental design and performance of a typical flow cytometric assay aimed to detecting intracellular ROS generation induced by an exogenous prooxidant chemical, such as plumbagin, a redox cycling compound generating superoxide anion radical. Graphs in row (**A**) show the strategy for observation of live and dead Jurkat cells by their scatter properties (Graph A1, FS INT Lin vs. SS INT Lin) and for selection of single cells (Graph A2 FS Peak vs. FS INT dotplot). In graph A3, single cells either live (fuchsia) or dead (green) are confirmed and selected by using the viability stain DAPI. On the basis of live cells, panels (**B**,**C**) show the performance of range-finding experiments to define plumbagin cytotoxicity (panel **B**) and for titrating the fluorescent probe MitoSOX Red (panel **C**), selected for assessing $O_2^{\bullet-}$ generation by plumbagin. Plumbagin is relatively low-toxic for Jurkat cells (panel **B**), and the concentration 80 μM is chosen for titrating MitoSOX Red (panel **C**). By incubating Jurkat cells at a fixed cell density (100,000 cells/mL) with a series of Mitosox Red concentrations in the presence of 80 μM plumbagin, it is shown that over 500 nm, Mitosox Red no significant increase in fluorescence intensity is observed, and thus this dye concentration (and dye-to-cell ratio) is selected for the toxicity examples shown in panels (**D**,**E**). A similar titration procedure is followed for the probes HE and $H_2$DCF-DA used in these examples. Graphs in row (**D**) (control experiment) show that the mitochondrial probe MitoSOX Red (D4) allows a much better discrimination of intracellular ROS levels between live (fuchsia) and dead (green) cells than its related probe HE (D5). Both orange-emitting probes have preferential sensitivity towards $O_2^{\bullet-}$ anion. Cytosolic ROS-sensitive probe $H_2$DCF-DA (D6) has an intermediate efficiency in separating ROS levels in live or dead cells in control conditions, as reflected by the green fluorescence emission of DCF. Graphs in row E, however, show a similar efficiency between MitoSOX Red (E7) and HE (E8) for detecting plumbagin-induced ROS generation. Additionally, $H_2$DCF-DA (E9) is able to respond, albeit at lower efficiency, to the incubation with the redox cycling compound plumbagin, supporting the relative lack of specificity of this probe usually considered as a sensor of $H_2O_2$ or peroxidative activity.

Figure 9A shows the strategy followed by us to identify single live cells, separated from debris and cell doublets (or aggregates). The scatter dotplot (FS Integral vs. SS Integral) shows two populations of events, likely including cells and debris. A second scatter dotplot (FS Peak vs. FS Integral) 2 shows the gate allowing to discriminate single cells from doublets/aggregates. The identification of live cells is confirmed with the vital probe DAPI, which stains live cells (panel 3) and with FS characteristics, by identifying dead or dying cells. Accordingly, all the analyses related to the use of ROS-sensitive fluorescent probes in

FCM studies, including range-finding and titration experiments should be performed on events included simultaneously in the gates of "live cells" and "single cells".

Finally, and based upon our own experience and the experimental evidences discussed in this review, in Table 2 we propose some recommendations for performing in vitro studies of oxygen and oxygen-related stress by FCM. Recommendations include the selection of the more appropriate fluorescent reagents and of suitable positive biological controls as well as for the inclusion of viability markers in the experiments. Such recommendations are aimed to help to standardize and to improve the specificity and sensitivity of RONS analysis by FCM.

**Table 2.** Some recommendations for performing in vitro functional studies of oxygen and oxygen-related stress by flow cytometry.

| | Biological Process of Interest and Experimental In Vitro Setting | | | | | |
|---|---|---|---|---|---|---|
| | Peroxidative Activity | | | Redox Cycling | | Antioxidant Defense |
| Probe | Mito-PY1 | $H_2$DCF-DA | DHRH123 | MitoSOX Red | HE | Monochloro-bimane |
| Viability Stain | DAPI or PI | DAPI or PI | DAPI or PI | DAPI | DAPI | PI |
| Positive control | $H_2O_2$ | t-BOOH | CHP | Menadione | Plumbagin | N-Acetyl cysteine GSH-ester |

**Author Contributions:** Conceptualization, J.-E.O.; investigation, B.J., G.H., A.M.-R. and J.-E.O.; writing—original draft preparation, B.J., G.H., A.M.-R. and J.-E.O.; writing—review and editing, B.J., G.H., A.M.-R. and J.-E.O.; supervision, J.-E.O. All authors have read and agreed to the published version of the manuscript.

**Funding:** This research received no external funding.

**Institutional Review Board Statement:** Not applicable.

**Informed Consent Statement:** Not applicable.

**Data Availability Statement:** Examples of listmode data files supporting reported results can be found at Flow Repository: http://flowrepository.org/id/FR-FCM-Z6DP (accessed on 20 May 2023).

**Conflicts of Interest:** The authors declare no conflict of interest.

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
