# Peer review of "Flow Cytometry of Oxygen and Oxygen-Related Cellular Stress"

_oxygen, doi:10.3390/oxygen3020016_

Round 1

Reviewer 1 Report

In this review paper is reported various chemicals to detect local levels of oxygen and reactive oxygen species and their application to flow cytometry to evaluate oxidative stress in living cells.

1. The reactions of ROS or oxygen and the fluorescent dyes compete with physiological in situ reactions. This is a great limitation for the methodology. The consumption by the dyes should be measured. This should be discussed somewhere.

2. RONS is not familiar to all the readers and should be spelled out when it appears for the first time.

3. page 2, line 73-page 3

Diffusion of the dye across the plasma membrane into the cell and across the mitochondrial membrane into the mitochondria is a big issue in this research field. This should be independent of the Introduction (1. Introduction: Oxygen, ROS and Oxidative Stress). 

4. Page 3, line 93.  Proton-motive force (   m) ???

5. The authors should be careful on mistakes on subscript, superscript, and expression of radicals, such as line 387, line 420, line 817, etc.

6. Page 9-Page 14. So many nomenclatures and trade names are seen in these sections. It would be helpful for the readers to show the list of chemical structure for each chemical, as shown in Figure 2. Otherwise, it is hard to follow the texts over several pages. Please elaborate on this point. The revision will make this review paper very meaningful.

Author Response

Reviewer1:

We thank the reviewer for her/his worth comments and suggestions for our manuscript.

In this review paper is reported various chemicals to detect local levels of oxygen and reactive oxygen species and their application to flow cytometry to evaluate oxidative stress in living cells.

  1. The reactions of ROS or oxygen and the fluorescent dyes compete with physiological in situ reactions. This is a great limitation for the methodology. The consumption by the dyes should be measured. This should be discussed somewhere.

The reviewer is right in his/her oportune comment. Accordingly, we have rewritten the end of the section 1 as follows (page 2, lines 81-90):

“However, fluorescence-based detection of ROS is a complex task due to the low concentration, the short half-life of most ROS and the extensive interactions among ROS, as well as to the competing effect of the exogenous fluorescent probes with the physiological in situ reactions. All these issues are at the basis of many intrinsic limitations of both probes and experimental conditions [51-53], and its contribution should be assessed. In addition, the efficiency and specificity of many probes for detecting ROS in vitro still need to be established [51-53]. Such limitations and potential sources of artifacts complicate quantitative measurements of intracellular generation of ROS and demand careful design of the experiments and cautious interpretation of the results [53].”

In addition, we have discussed this issue also  in the Section 5.2. Choice of Fluorescent Probes (page 24, lines 1160-1167), as follows:

“Thus, while the different fluorescent probes responded individually to the addition of pro-oxidant compounds as expected, the probes in spectrally-compatible combination with other probes showed important increases or reductions in the fluores-cence generation rate induced by the same prooxidant. Such observations are compatible with the consumption or generation of ROS by some of the probes in the tested pairs.”

  1. RONS is not familiar to all the readers and should be spelled out when it appears for the first time.

The term RONS has been spelled out as Reactive Oxygen and Nitrogen Species (RONS) [7] in page 2, lines 51-52 .

  1. page 2, line 73-page 3. Diffusion of the dye across the plasma membrane into the cell and across the mitochondrial membrane into the mitochondria is a big issue in this research field. This should be independent of the Introduction (1. Introduction: Oxygen, ROS and Oxidative Stress).

We agree with the reviewer. To meet this request, we have moved Figures 1 and 2 from Section 1 (Introduction: Oxygen, ROS and Oxidative Stress) to Section 3 (General Strategies in Flow Cytometric Analysis of Oxygen and Oxidative Stress).  In that Section (page 6, lines 295-301) the point “c) Direct detection of ROS, the initiators of the oxidative stress process” has been rewritten as follows: “This task is complex due to the low concentration, short half-life and extensive inter-actions of ROS, as well as by the limitations imposed by the fluorescent probes and the experimental conditions [53]. As schematically shown in Fig. 1 and Fig. 2, typically, the fluorescent probes are non fluorescent until being oxidised by intracellular oxidants and they are incorporated in form of fluorogenic substrates (Fig. 1) which have been modified by appropriate chemical design to become both cell-permeable and susceptible to ROS-mediated oxidation (Fig. 2) [51-53]”.

In addition, we have added a new paragraph in Section  4.3. Influence of the Probes on the Experimental System (page 19, lines 851-858), as follows:

“Another aspect to take into account in the use of probes with mitochondrial localization, such as MitoPY1,  DHRH123 or MitoSox Red is the possible effect of such probes on the mitochondrial membrane potential. Mitochondrion-targetted chemical probes are molecules with cationic groups (triphenylphosphonium for MitoSox Red and MitoPY1; imino group for DHRH123) that are attracted to the negative electric charge generated by mitochondrial respiration. An excess of probe input of positive charge could reduce the net negative charge of the mitochondria and decrease the uptake of new cationic molecules.”

To make those changes more clear and meaningful, we have renamed Section 2 as “2. Flow Cytometry as a Tool for Functional Cell Research”.

  1. Page 3, line 93. Proton-motive force ( m) ???

We thank the reviewer for noticing this typo. The symbol for proton-motive force (Δψm) has been corrected (currently on page 6, line 314)

  1. The authors should be careful on mistakes on subscript, superscript, and expression of radicals, such as line 387, line 420, line 817, etc.

We thank the reviewer for the warning and we apologize for the mistakes. We have checked thoroughly the manuscript for those and other typing errors.

  1. Page 9-Page 14. So many nomenclatures and trade names are seen in these sections. It would be helpful for the readers to show the list of chemical structure for each chemical, as shown in Figure 2. Otherwise, it is hard to follow the texts over several pages. Please elaborate on this point. The revision will make this review paper very meaningful.

We thank the reviewer for this excelent suggestion. To meet this task, we have incorporated five new figures, showing the publicly-available chemical structures of fluorescent probes. For generating these new figures, we have used MolView, an open source chemical modeling package (http://molview.org/). The chemical structures have been retrieved via MolView from the PubChem Compounds database or, when unavailable, sketched by us with MolView from published structures. In that way, most of the fluorescent probes mentioned in this review (and, of course, the most used) are identified by their chemical name or their tradename, with the abbreviation in parenthesis. The new figures are:

Figure 4. Chemical structures of relevant fluorescent probes used to monitor intracellular oxygen in hypoxic conditions. (page 10, line 424).

Figure 5. Chemical structures of relevant fluorescent probes used to monitor singlet oxigen. (page 12, line 491).

Figure 6. Chemical structures of relevant fluorescent probes used to monitor Reactive Oxygen Species. (page 14, line 632).

Figure 7. Chemical structures of relevant fluorescent probes used to detect lipid peroxidation. (page 16, line 721).

Figure 8. Chemical structures of relevant fluorescent probes used to assess antioxidant defenses. (page 18, line 786).

Reviewer 2 Report

The comprehensive review on the detection and quantification of reactive oxygen and nitrogen species (ROS and RNS) using flow cytometry (FCM) is praiseworthy.

Comprehensiveness: The authors have done an excellent job providing a thorough discussion on the methodologies for the detection and analysis of ROS and RNS. They have covered various fluorescent probes, fluorogenic substrates, and other detection methods, along with the technical details of optimizing these approaches. This provides readers with a holistic view to understand this complex field.

Depth: The manuscript goes into great depth when discussing various fluorescent probes. It explains the pros and cons of different probes and how to choose the one most appropriate for a specific research objective. This depth is appreciated.

Recommendations: The authors have provided detailed recommendations for conducting FCM studies, including the selection of suitable fluorescent reagents and biological positive controls, and the inclusion of viability markers in the experiments. These recommendations are practical and can help laboratory workers to improve the specificity and sensitivity of their experiments.

Overall, this manuscript is a substantial contribution to the field of oxidative stress research. The comprehensiveness and depth of the manuscript are remarkable, despite the inclusion of a vast amount of technical details. I recommend it for publication after considering the minor revisions.

Author Response

Reviewer 2:

This reviewer has not concerns or questions about our manuscript. We thank him/her for the kind comments about our review.

Reviewer 3 Report

It is a well-written manuscript, focused on the specific topic of cytometric studies of oxidative stress. Flow cytometry represents the most rapid and powerful tool for investigating ROS at the single-cell level with high sensitivity and reproducibility. Overall, this is a well-organized, and detailed manuscript. The review is well-presented and follows a logical flow. I have only a few comments.

1. All abbreviations used in the text should be defined on first use (e.g. DHRH123)

2. CellROX (lines 473-482): The authors could present the limitations of this method.

3. I noticed about 60 literature sources published 20 years ago and even from the previous century. For faster-paced fields such as flow cytometry, sources published in the last 5 years are a good reference, as these sources are more up-to-date and reflect the latest processes or best practices. Table 1 could also be updated. The authors refer to the work of 2001.

Author Response

Reviewer 3:

We thank the reviewer for his/her worth suggestions and comments. We thank him/her for the kind comments about our review.

It is a well-written manuscript, focused on the specific topic of cytometric studies of oxidative stress. Flow cytometry represents the most rapid and powerful tool for investigating ROS at the single-cell level with high sensitivity and reproducibility. Overall, this is a well-organized, and detailed manuscript. The review is well-presented and follows a logical flow. I have only a few comments.

  1. All abbreviations used in the text should be defined on first use (e.g. DHRH123).

We have checked thoroughly the manuscript and corrected this deficiency.

  1. CellROX (lines 473-482): The authors could present the limitations of this method.

To meet this request, we have incorporated  a new paragraph in the section “5.2. Choice of Fluorescent Probes”, as follows:

“The results obtained in our systematic studies with the CellROX® and the ROS-ID® reagents families of products [126, 196] have shown that, in general, all these probes present medium to low intensity in the responses to the different prooxidants tested, with a very broad specificity towards them, which is in accordance with the manufacturers' warnings, included in the documentation of the reagents and kits.”

  1. I noticed about 60 literature sources published 20 years ago and even from the previous century. For faster-paced fields such as flow cytometry, sources published in the last 5 years are a good reference, as these sources are more up-to-date and reflect the latest processes or best practices.

I agree with the reviewer in that flow cytometry  (FCM) is a very fast-paced area. In my opinion, this is mostly due to the sophisticated engineering improvements to the instruments and the computing- and data-mining software developments that allow state-of-the-art flow cytometers to determine routinely 10 to 30 and more fluorescent parameters simultaneously in each single cell of highly complex populations, and to extract meaningful biological information. This has been duly referenced in our review [refs. 54 to 57]. Such technological potential of FCM and its developments are mostly fulfilled in the fields of Immunology and Onco-Hematology, where FCM has become arguably an irreplaceable analytical tool (see our refs. 58, 59). However, this review is not aimed to present the technology behind FCM for oxidative stress studies. In fact, most of the basic cell biology studies in FCM still use simpler cytometers (or exploit only in part complex instruments) with a limited number of fluorescent biomarkers examined simultaneously. The final goal of this review is to contribute (with my 40-year long experience in the area) to help basic scientists, and most especially, newcomers, in being aware of the complications, limitations and caveats in such studies, in order to apply properly FCM to oxidative stress research. Therefore, in the original submission we had included older references to illustrate the work with classical fluorescent probes and their limitations. 

To meet the reviewer request, in the current version we have gone extensively trough the original reference list. We have updated 65 references to include much more recent examples of the development and applications of classical and novel fluorescent reagents. We have kept older references illustrating the complications and limitations of the fluorescent probes, as in many cases, such caveats were issued many years ago. In our own experience, we find that still many users of such reagents are unaware of this important aspect, which may lead to non-adequate design of the experiments or to an incorrect interpretation of the results.

Table 1 could also be updated. The authors refer to the work of 2001.

The citation to Table 1 was wrong in the original review. The right citation should be [53], which is dated on 2017.

Round 2

Reviewer 1 Report

The manuscript is revised well according to the comments from the reviewer.